# Structural insight into mitochondrial β-barrel outer membrane protein biogenesis

Kathryn A. Diederichs[1,5], Xiaodan Ni[2,5], Sarah E. Rollauer[1,3,4,5], Istvan Botos [1], Xiaofeng Tan[2], Martin S. King[3], Edmund R. S. Kunji [3], Jiansen Jiang [2✉] & Susan K. Buchanan [1✉]

In mitochondria, β-barrel outer membrane proteins mediate protein import, metabolite transport, lipid transport, and biogenesis. The Sorting and Assembly Machinery (SAM) complex consists of three proteins that assemble as a 1:1:1 complex to fold β-barrel proteins and insert them into the mitochondrial outer membrane. We report cryoEM structures of the SAM complex from *Myceliophthora thermophila*, which show that Sam50 forms a 16-stranded transmembrane β-barrel with a single polypeptide-transport-associated (POTRA) domain extending into the intermembrane space. Sam35 and Sam37 are located on the cytosolic side of the outer membrane, with Sam35 capping Sam50, and Sam37 interacting extensively with Sam35. Sam35 and Sam37 each adopt a GST-like fold, with no functional, structural, or sequence similarity to their bacterial counterparts. Structural analysis shows how the Sam50 β-barrel opens a lateral gate to accommodate its substrates.

[1] Laboratory of Molecular Biology, National Institute of Diabetes & Digestive & Kidney Diseases, National Institutes of Health, 9000 Rockville Pike, Bethesda, MD 20892, USA. [2] Laboratory of Membrane Proteins and Structural Biology, National Heart, Lung, and Blood Institute, National Institutes of Health, Bethesda, MD 20892, USA. [3] MRC Mitochondrial Biology Unit, University of Cambridge, Cambridge Biomedical Campus, Cambridge CB2 0XY, UK. [4]Present address: Vertex Pharmaceuticals, 50 Northern Avenue, Boston, MA 02210, USA. [5]These authors contributed equally: Kathryn A. Diederichs, Xiaodan Ni, Sarah E. Rollauer. ✉email: jiansen.jiang@nih.gov; susan.buchanan2@nih.gov

β-barrel membrane proteins are found in the outer membranes of mitochondria, chloroplasts, and Gram-negative bacteria. Mitochondrial β-barrel membrane proteins are synthesized in the cytosol and imported across the mitochondrial outer membrane as unfolded precursor proteins by the translocase of the outer membrane (TOM) complex. In the intermembrane space, β-barrel precursor proteins interact with small translocase of the inner membrane (TIM) chaperones prior to their transfer to the sorting and assembly machinery (SAM) complex for folding and insertion into the outer membrane[1] (Fig. 1). The SAM complex consists of three components: a β-barrel core, Sam50, which spans the outer membrane, and two accessory subunits, Sam35 and Sam37, that associate with Sam50 on the cytosolic side of the membrane[2–6] (Supplementary Fig. 1). Sam50 and Sam35 are essential proteins, with Sam50 folding and inserting β-barrel substrates into the outer membrane and Sam35 interacting with the substrate β-signal located in the last β-strand[3]. Sam37, while not essential, functions in substrate release[3,7] and may also promote formation of a TOM–SAM supercomplex[8].

Bacterial β-barrel membrane proteins take a different pathway to reach the outer membrane, but they are inserted by evolutionarily related machinery[9] (Supplementary Fig. 1). In bacteria, proteins are synthesized in the cytoplasm, secreted across the inner membrane by the Sec translocon, bound to chaperones in the periplasm, and transferred to the bacterial assembly machinery (BAM) complex for folding and insertion into the outer membrane[10,11]. In *Escherichia coli*, four lipoproteins, BamB, BamC, BamD, and BamE, associate with the periplasmic domain of BamA (itself a 16-stranded transmembrane β-barrel) to fold and insert β-barrels ranging in size and complexity from 8 β-strands with a simple barrel fold, to 26 β-strands with multiple domains[12]. Structures of BamA and BAM complexes illustrate two features of BamA that facilitate folding and insertion: (1) BamA has a narrowed hydrophobic surface where the first and last β-strands meet, which locally compresses and destabilizes the lipid bilayer to allow membrane protein insertion (BamA assisted model). (2) The first and last β-strands form a "lateral gate" that has been shown to open and close by molecular dynamics (MD) simulations, disulfide crosslinking, and structures solved by X-ray crystallography and cryo-electron microscopy (cryoEM)[13–20]. The required opening and closing of the lateral

gate gave rise to the budding model, where a β-hairpin (or larger portion) of the substrate enters the BamA lumen, associates with unpaired β-strands on BamA, and uses the BamA β-barrel to sequentially fold into the outer membrane, budding off from BamA when the first and last strands of the substrate come together. With the wide variety and large quantities of β-barrel proteins that exist in bacteria, it is possible that either or both models, or variations of these[21,22], are used depending on the complexity of the substrate.

Sam50 and BamA are evolutionarily related, and both are members of the Omp85 superfamily[9,23]. Although no structures had previously been determined for any of the SAM components, structures exist for Omp85 family members BamA, TamA, and FhaC[13–18,20,24–27]. A homology model based on BamA was used to study mitochondrial membrane protein insertion by crosslinking precursor proteins to predicted strands β1 and β16 of Sam50, suggesting that substrates enter the lumen of Sam50, accumulate at the lateral gate, and are released into the compressed membrane adjacent to the gate[28]. In contrast to bacteria, mitochondria are predicted to make only four types of β-barrel outer membrane protein: Sam50, Tom40, VDAC, and Mdm10[29]. Structures of Tom40[30,31] and VDAC[32] reveal very similar 19-stranded β-barrel proteins; Mdm10 is expected to adopt the same fold. Therefore, it appears that Sam50 substrates (other than Sam50 itself) may use a single folding and insertion mechanism.

Although evolutionarily related, a number of important differences between SAM and BAM exist (Supplementary Fig. 1), suggesting that the details of how substrates are targeted to the folding machinery, where and how those substrates interact with peripheral SAM and BAM components, the functions of those peripheral components, and how substrates are released into the membrane once folded, will differ substantially. First, the entry pathways clearly differ as described above. Mitochondrial proteins enter from outside the organelle but interact with Sam50 on the intermembrane space side of the outer membrane. In contrast, bacterial substrates in the periplasm associate with periplasmic lipoproteins BamB, BamC, BamD, and BamE. BamD is essential in this process and has been shown to interact with unfolded substrates, activating BamA for substrate folding[33]. The peripheral SAM components, Sam35 and Sam37, sit on the opposite side of the membrane and therefore cannot make the

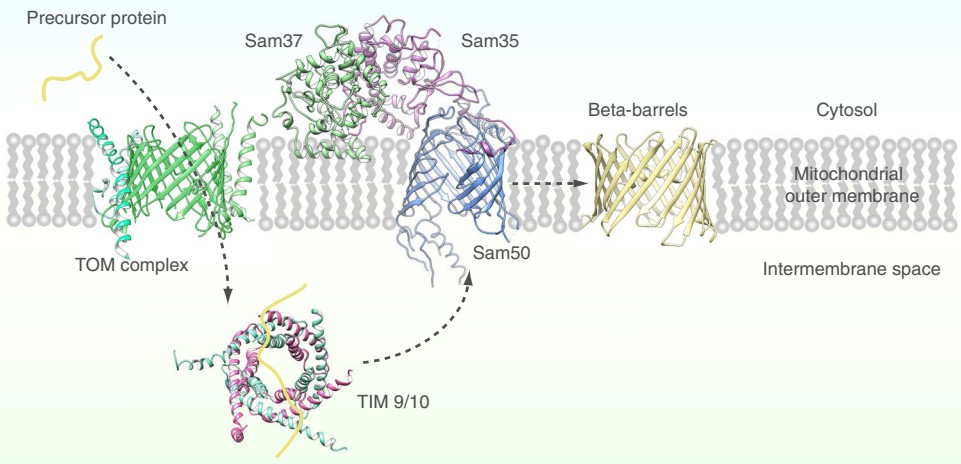

**Fig. 1 Schematic of β-barrel outer membrane protein biogenesis in mitochondria.** Mitochondrial outer membrane β-barrel proteins are synthesized in the cytosol (yellow), then imported into the intermembrane space of the mitochondria by the translocase of the outer membrane complex (TOM complex, green). Once in the intermembrane space, the precursor protein is bound by chaperone proteins (TIM9/10, magenta/cyan) and directed to the sorting and assembly machinery complex (SAM complex, blue/light green/orchid). The SAM complex facilitates outer membrane β-barrel precursor protein folding and insertion into the outer membrane.

same types of interactions with substrates that BAM lipoproteins do. Second, mitochondria contain only one POTRA domain while bacteria contain multiple POTRA domains[23]. In bacteria, the POTRA domain nearest the β-barrel is essential for activity[34], while in mitochondria it can be removed without major consequence[3]. Third, mitochondrial Sam35 and Sam37 have been demonstrated to have functions not found in BAM lipoproteins[7].

To better understand how β-barrel proteins are folded in mitochondria, we used cryoEM to solve structures of the complete SAM complex in lipid nanodiscs at 3.4 Å resolution, and in detergent to 3.0 Å resolution, with six structures in total (one in lipid, five in detergent). The SAM complex consists of one copy each of Sam50, which spans the mitochondrial outer membrane, and Sam35 and Sam37, which sit on the cytosolic side of the membrane. Sam50 forms a 16-stranded β-barrel which is minimally closed between strands β1 and β16. The single POTRA domain extends into the intermembrane space away from the barrel lumen, potentially allowing substrate access from this side of the membrane. The N-terminal portion of Sam35 interacts extensively with Sam50 on the cytosolic side of the membrane, occluding substrate efflux. Sam35 also interacts extensively with Sam37, such that Sam37 makes no contacts with Sam50 on the cytosolic side of the membrane; however, the linker between two predicted transmembrane α-helices in Sam37 interacts with the Sam50 POTRA domain in the intermembrane space. A comparison of the structures shows how Sam50 opens a lateral gate to fold and insert substrates.

## Results

**Preparation of SAM complexes**. To obtain a homogeneous SAM complex, we co-expressed *M. thermophila* (recently renamed to *Thermothelomyces thermophilus*) Sam50, Sam35, and Sam37 in *Saccharomyces cerevisiae* (see Methods for details). The SAM complex was solubilized from isolated mitochondria using the detergent lauryl maltose neopentyl glycol (LMNG) and purified by affinity chromatography using a Twin-Strep tag on the N-terminus of Sam37. Before incorporating into lipid nanodiscs, the SAM complex was further purified on a size-exclusion column using LMNG (Supplementary Fig. 2A–D). For single particle analysis in detergent, LMNG was exchanged on the size-exclusion column for glycol-diosgenin (GDN), a synthetic substitute for digitonin (Supplementary Fig. 2E, F). The purified samples contained approximately stoichiometric amounts of each of the three subunits.

**The SAM complex is monomeric in lipid nanodiscs**. The cryoEM structure of the SAM complex in lipid nanodiscs was reconstructed from 179,509 particles and yielded a 3.4 Å resolution structure (Fig. 2, Supplementary Fig. 3 and Table 1). The resolution was sufficient to allow ab initio tracing of a majority of the folds of all three subunits. In lipid nanodiscs, the SAM complex is monomeric and contains one copy each of Sam50, Sam35, and Sam37. Sam50 forms a 16-stranded transmembrane β-barrel with a single POTRA domain extending into the

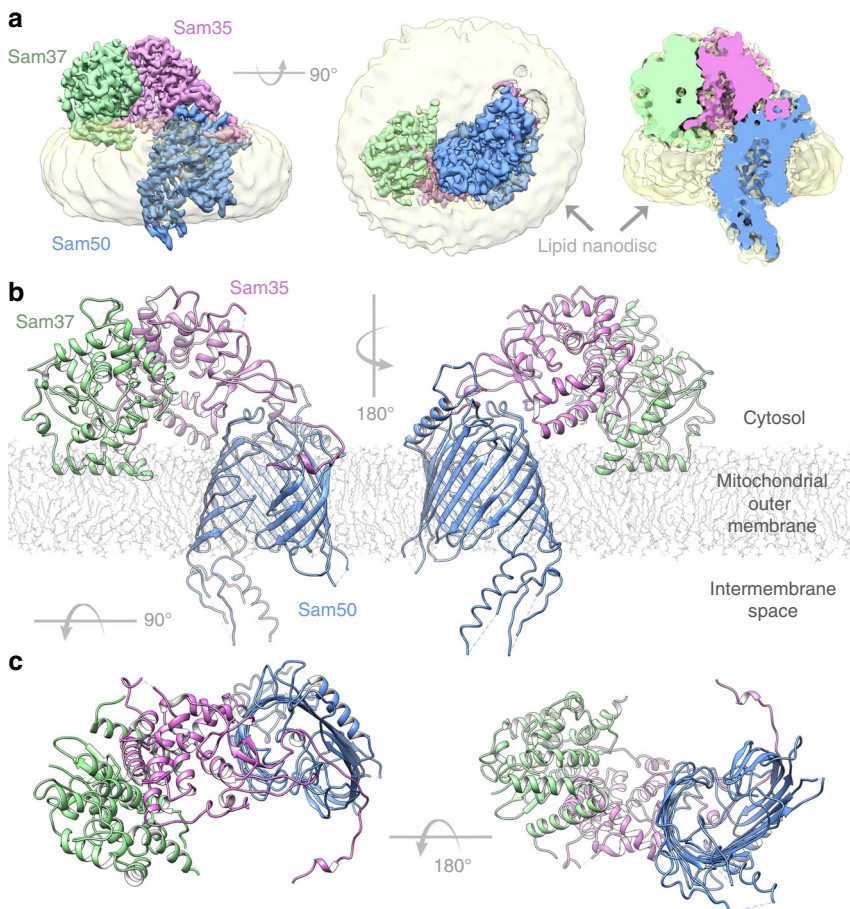

**Fig. 2 Structure of the SAM complex in lipid nanodiscs. a** The cryoEM density map of the SAM complex (Sam35 in orchid, Sam37 in pale green, and Sam50 in blue) with the nanodisc densities (transparent yellow) showing the membrane boundaries. A cross section is shown on the right. **b** Ribbon views of the SAM complex in the context of a model lipid bilayer. **c** Ribbon views from top (cytosol) and bottom (intermembrane space). See also Supplementary Fig. 3 and Table 1.

**Table 1 CryoEM data collection, structure determination and model statistics.**

| | SAM in lipid nanodiscs | | | SAM in detergent | | |
|---|---|---|---|---|---|---|
| *Data collection* | | | | | | |
| Nominal magnification | ×130,000 | | | ×130,000 | | |
| Voltage (kV) | 300 | | | 300 | | |
| Exposure time (s/frame) | 0.2 | | | 0.2 | | |
| Number of frames | 50 | | | 50 | | |
| Total dose (e-/Å$^2$) | 69 | | | 69 | | |
| Defocus range (μm) | −0.8 to −3.0 | | | −0.8 to −3.0 | | |
| Pixel size (Å) | 1.06 | | | 1.06 | | |
| *Image processing* | | | | | | |
| Micrographs selected | 11,347 | | | 10,831 | | |
| | | **Dimer 1** | **Monomer from Dimer 1** | **Dimer 2** | **Dimer 3** | **Monomer** |
| Initial particle images (no.) | 3,951,406 | 174,217 | – | 112,106 | 380,304 | 514,800 |
| Final particle images (no.) | 179,509 | 117,339 | 335,670 | 60,472 | 122,361 | 138,575 |
| Symmetry imposed | C1 | C2 | C1 | C2 | C2 | C1 |
| FSC threshold | 0.143 | 0.143 | 0.143 | 0.143 | 0.143 | 0.143 |
| Final map resolution (Å) | 3.4 | 3.2 | 3.0 | 3.6 | 3.9 | 3.7 |
| *Atomic model* | | | | | | |
| Number of protein residues | 1037 | 2190 | 1087 | 2192 | 1326 | 887 |
| *Validation* | | | | | | |
| Most favored (%) | 87.51 | 87.21 | 93.8 | 83.49 | 81.6 | 83.93 |
| Allowed (%) | 12.49 | 12.7 | 6.2 | 16.42 | 18.4 | 16.07 |
| Disallowed (%) | 0 | 0.09 | 0 | 0.09 | 0 | 0 |
| Rotamer outliers (%) | 5.42 | 6.92 | 5.79 | 1.06 | 1.14 | 0 |
| r.m.s.d Bond lengths (Å) | 0.003 | 0.005 | 0.008 | 0.008 | 0.005 | 0.007 |
| r.m.s.d Bond angles (°) | 0.651 | 0.812 | 0.876 | 0.922 | 0.79 | 0.887 |
| Clashscore | 8.96 | 7.98 | 3.66 | 10.08 | 13.92 | 9.69 |
| Map CC (main chain) | 0.79 | 0.8 | 0.83 | 0.81 | 0.79 | 0.81 |
| Map CC (side chain) | 0.77 | 0.78 | 0.81 | 0.8 | 0.78 | 0.79 |
| *Deposition ID* | | | | | | |
| EMDB | EMD-21913 | EMD-21915 | EMD-21918 | EMD-21916 | EMD-21917 | EMD-21914 |
| PDB | 6WUH | 6WUL | 6WUT | 6WUM | 6WUN | 6WUJ |

intermembrane space (Supplementary Fig. 4). Sam35 and Sam37 are located on the cytosolic side of the membrane, where Sam35 caps the barrel lumen of Sam50 and also interacts extensively with Sam37 (Fig. 2b, c). The orientations of Sam35 and Sam37, as well as visualization of the lipid nanodisc, suggest that Sam35 and Sam37 interact peripherally with the outer leaflet of the membrane (Fig. 2a) and residues 339–353 of Sam37 form an amphipathic α-helix that appears to interact with the membrane (Fig. 2b). In this structure, the highest resolution is observed for the Sam50 β-barrel and for Sam35, with lower resolution (and more flexibility) observed for Sam37 and the Sam50 POTRA domain (Supplementary Fig. 3D). When the map is filtered to low resolution, the first of two predicted transmembrane α-helices in Sam37 is visible; however, it is not visible when filtered to high resolution (Supplementary Fig. 3E). It appears that the relatively large size of the lipid nanodisc compared to the Sam50 β-barrel prevents optimal particle alignment and further resolution improvement.

**The SAM complex forms dimers in detergent.** In addition to using lipid nanodiscs, we determined the SAM complex structure in the detergent GDN using cryoEM (see Methods for details). In this sample, 3D reconstructions exhibited several conformations, with monomer and multiple dimer conformations present (here, the term monomer refers to the SAM complex, with one copy each of Sam50, Sam35, and Sam37; the dimer refers to two SAM complexes) (Fig. 3, Supplementary Figs. 5–7 and Table 1). The monomer conformation, although determined to a resolution of only 3.7 Å, is virtually identical to the monomer determined in lipid nanodiscs and to the monomer derived from dimer 1, described below. All of the dimers appear to be non-physiological,

with an up-down association of SAM complexes not anticipated from functional studies. However, we were able to determine a structure of the dimer with the most homogeneous conformation (dimer 1) at 3.2 Å resolution from 117,339 particles (Fig. 3a). The SAM complex monomer determined from lipid nanodiscs was fitted into half of dimer 1, revealing that they are in an almost identical conformation (Fig. 3b). Therefore, we performed particle symmetry expansion in dimer 1, focused on only half of the dimer, and proceeded to refine this monomer to a final resolution of 3.0 Å, resulting in more clearly resolved side chain densities (Supplementary Figs. 5, 6). The first transmembrane α-helix of Sam37 is well defined, spanning the membrane adjacent to the Sam50 β-barrel, but making no contact with the β-barrel. This contrasts with the recently reported high-resolution structures of the TOM core complex, where transmembrane α-helices closely associate with the hydrophobic surface of the Tom40 β-barrel[30,31]. The linker that connects the two predicted transmembrane helices in Sam37 interacts with the Sam50 POTRA domain, contributing an additional antiparallel β-strand to that domain.

**The Sam50 β-barrel is only partially closed.** As had been predicted from homology to BamA[20], Sam50 consists of a 16-stranded transmembrane β-barrel preceded by a single POTRA domain in the intermembrane space (Fig. 3c, d and Supplementary Fig. 9). The POTRA domain adopts the classic βααββ fold, and is positioned away from the barrel lumen, allowing substrate access in this conformation. Structures of Omp85 proteins illustrate how mobile POTRA domains can be, potentially allowing or blocking access to the barrel lumen from the periplasm (bacteria) or intermembrane space (mitochondria). The

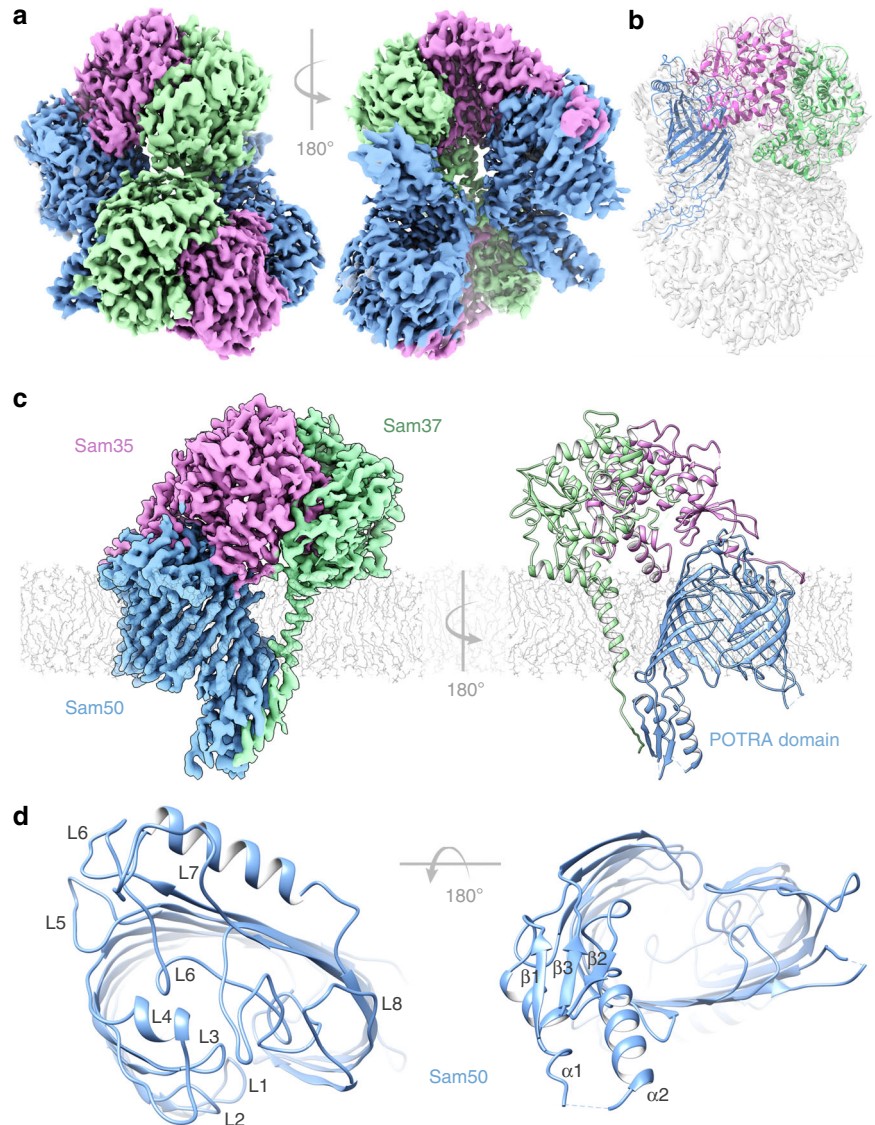

**Fig. 3 Structure of the SAM complex in detergent GDN. a** The cryoEM density map of a dimer of the SAM complex (dimer 1) in GDN with Sam 35 in orchid, Sam37 in pale green and Sam50 in blue. **b** Superposition of the structure of the SAM complex in nanodiscs (ribbons) to the cryoEM density map of dimer 1 (transparent surface). **c** Side view of the high-resolution structure of the SAM complex reconstructed from the dimer 1 particles using symmetry expansion. The cryoEM density map and the atomic model are shown on the left and right, respectively. **d** The top and bottom views of Sam50 barrel showing the cytosolic loops L1-L8 and the POTRA domain with a βααββ fold. See also Supplementary Figs. 5, 6, 7, and Table 1.

reduced resolution in this domain in the lipid nanodisc structure further illustrates its mobility (Supplementary Fig. 3D). On the cytosolic side of the membrane, Sam50 loops L1–L8 extend from the surface, positioning loops L4, L7, and L8 to interact with several N-terminal residues of Sam35. Loop 7 contains a surface-exposed α-helix that sits parallel to the membrane, while L6 extends deep into the empty barrel. These interactions effectively close the β-barrel on the cytosolic side, preventing substrate efflux. In both lipid and detergent, the Sam50 β-barrel adopts a kidney bean shape, with dimensions 50 by 40 Å (Figs. 2c, 3d and Supplementary Fig. 7). It is only partially closed at the interface, making no direct interactions between strands β1 and β16 (Supplementary Fig. 8). The short lengths of strands β1–β4 and β15–β16, and their orientation in the membrane, suggest that the lipid bilayer is likely to be compressed and disordered in this region, as has been observed for BamA[19,20]. We note that the conformation of the interface between strands β1 and β16 is virtually identical in lipid and detergent, allowing all subsequent

analyses to be carried out with the higher resolution detergent-based structure of Sam50 (Supplementary Fig. 7D).

**Sam35 and Sam37 adopt a GST-like fold**. Sam35 and Sam37 share no sequence similarity with any of the BAM lipo-proteins. The structures of Sam35 and Sam37 reveal that they each adopt a GST-like fold, which is also completely different from the BAM lipoproteins. Sam35 and Sam37 exhibit canonical N-terminal α/β domains and all α-helical C-terminal domains (Fig. 4a, b). Superposition of Sam35 and Sam37 shows the extensive structural similarities (Fig. 4c). Although Sam35 and Sam37 belong to the family of GST-like proteins, they do not have the necessary active site residues to catalyze conjugation of glutathione to a substrate (Fig. 4d). We compared the true GST protein (PDB:6J3F) with GST-like proteins (PDB:4IBP and PDB:3IC8), including Sam35 and Sam37 (Fig. 4e). Superpositions for Sam35 and GST, or for Sam37 and GST, have RMSDs of 2.5 Å

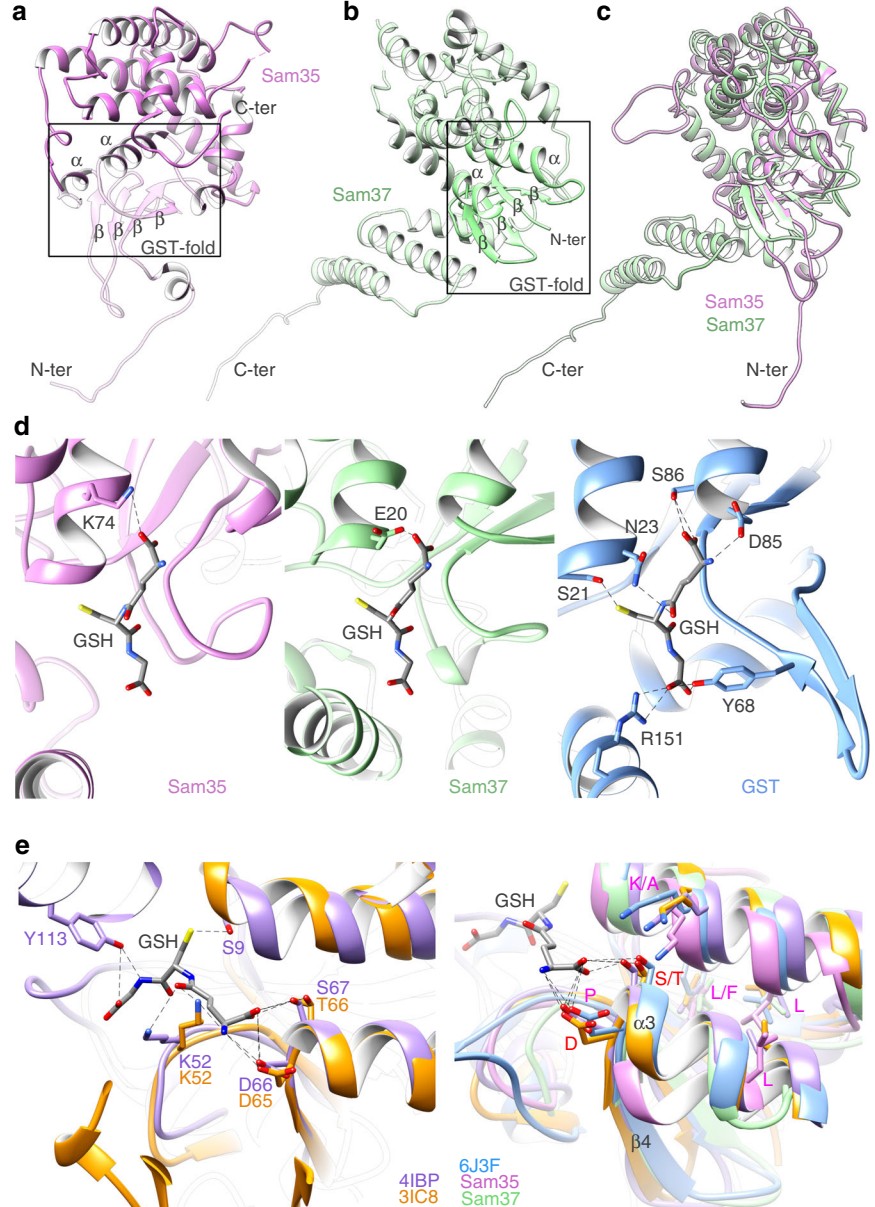

**Fig. 4 Analysis of the GST-like fold for Sam35 and Sam37. a** Ribbon view of Sam35 exhibits a canonical GST fold of an α/β domain at the N-terminus and an all α-helical C-terminal domain. **b** Ribbon view of Sam37 shows a configuration similar to Sam35 with an N-terminal α/β domain with an all α-helical C-terminal domain. **c** Superposition of Sam35 (orchid) and Sam37 (pale green) shows high structural similarity. **d** Close-up of the pseudo active site in Sam35, Sam37, and the genuine active site of GST (PDB:6J3F). Hydrogen bonds are shown as dashed lines. **e** Close-up views of the residues related to the active site of GST. Left: superposition of two GST-like proteins (left; PDB:4IBP and PDB:3IC8). Right: superposition of GST (PDB:6J3F), the same two GST-like proteins (PDB:4IBP and PDB:3IC8), Sam35 (orchid), and Sam37 (pale green). The bound GSH is shown in gray. The active site residues conserved among all the above structures are labeled in magenta, whereas residues only present in GST and GST-like proteins are labeled in red.

and 2.2 Å, respectively. Clearly, secondary structure elements align well but active site residues are missing. Members of the GST protein family exhibit extreme diversity in sequence, and a large proportion of them are of unknown function. Sam35 plays an essential role in the SAM complex, and has been shown to bind precursor protein[3]. However, the SAM complex structure does not currently offer any clues to this interaction. Sam37 is not essential, but has been shown to function later in the folding process, facilitating precursor release[7]. It is interesting that the Sam37 linker binds to the POTRA domain of Sam50 in the intermembrane space, since the POTRA domain has also been implicated in precursor release[35]. This interaction places the two subunits with a shared function in close proximity. Our structure

supports the previous observations that Sam37 serves to stabilize the interaction of Sam35 with Sam50, as there are extensive interactions between cytosolic Sam37 and Sam35[7,36].

**Subunit interactions between Sam50 and Sam35 and Sam37.** Sam35 interacts with a groove in Sam50 created by loops L4 and L7, primarily through residues near its N-terminus. The interactions consist of 15 hydrogen bonds and 3 salt bridges resulting in a buried surface area of 3918.6 Å$^2$ (Fig. 5a and Supplementary Tables 1, 2). These interactions allow Sam35 to sit tightly along the edge of the Sam50 β-barrel. In the intermembrane space, the Sam50 POTRA domain interacts with the linker connecting the

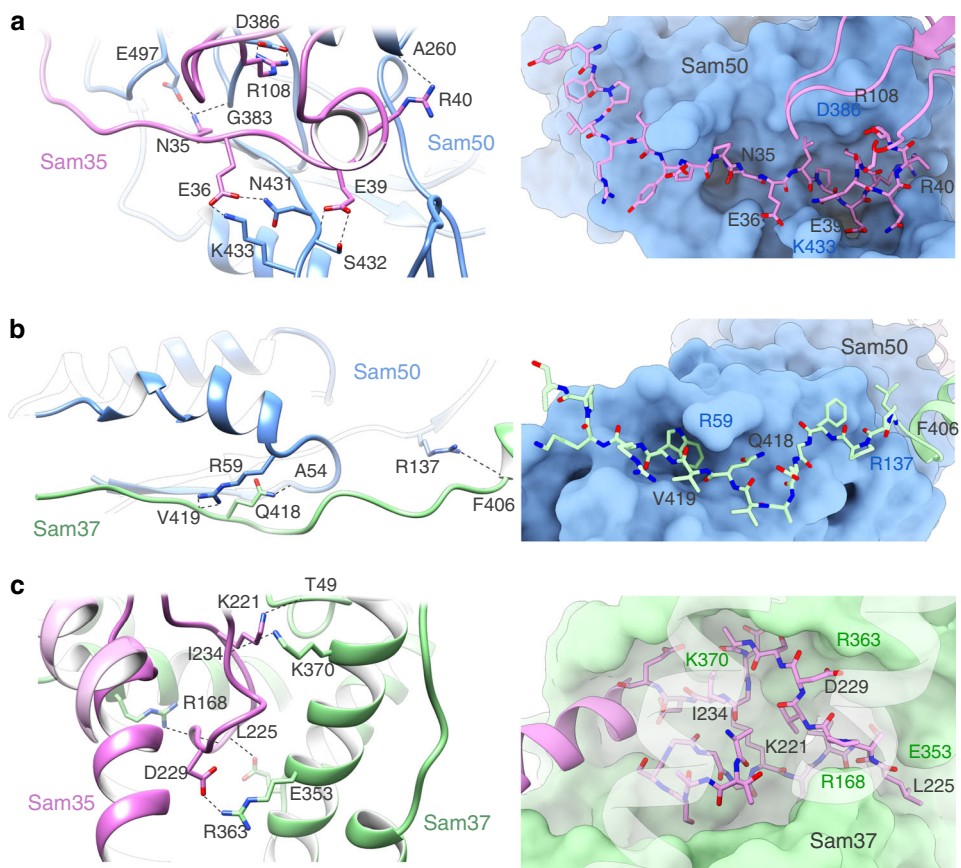

**Fig. 5 Interactions between individual components of the SAM complex.** Interfaces for (**a**) Sam35/Sam50 β-barrel, (**b**) Sam50 POTRA/Sam37, and (**c**) Sam35/Sam37, in cartoon and surface representations. The interacting residues are labeled. See also Supplementary Fig. 12 and Supplementary Tables 1, 2.

two predicted transmembrane helices of Sam37 through 8 hydrogen bonds, resulting in a buried surface area of 2789.8 Å$^2$ (Fig. 5b and Supplementary Tables 1, 2). Sam35 and Sam37 interact extensively on the cytosolic side of the membrane through 9 hydrogen bonds and 2 salt bridges, for a total buried surface area of 2978.5 Å$^2$ (Fig. 5c and Supplementary Tables 1, 2).

N-terminal truncation of Sam35 (Δ1–45) decreases Sam50 association with the ternary complex, as evaluated by a strep affinity pulldown assay (Supplementary Fig. 13A, B). The Sam35 N-terminal truncation disrupts the majority of residues identified in the Sam35–Sam50 interface, which supports the decrease in Sam50 assembly in the SAM complex observed. The small amount of Sam50 still present could be attributed to the additional residues in the Sam35–Sam50 interface that were not disrupted (S95, T92, T112) and the smaller Sam50–Sam37 interface which could allow for some Sam50 assembly into the SAM complex (Fig. 5 and Supplementary Tables 1, 2). The Sam35 N-terminal truncation did not substantially alter the stoichiometry of Sam35 and Sam37 in the ternary complex or Sam35 +Sam37 complex, which was expected due to the extensive Sam35–Sam37 interface that does not involve the N-terminus of Sam35 (Supplementary Fig. 13).

Of the conserved residues from structure-based sequence alignments (Supplementary Figs. 9–12 and Supplementary Table 3), only two interactions occur at the subunit interfaces involving highly conserved residues: Y156 in Sam35 with semi-conserved D111 on Sam37, and G383 in Sam50 with non-conserved N35 on Sam 35 (Supplementary Fig. 12 and Supplementary Table 3). Otherwise, there are no clear patterns of conserved residues in subunit interactions. In the detergent structure, the Sam37 linker between transmembrane domains

interacts with the Sam50 POTRA domain, but neither of these regions has high sequence conservation between species (Supplementary Fig. 12D). The assembly of the SAM complex was maintained when the whole POTRA domain was truncated (Δ1–135) (Supplementary Fig. 14), suggesting that this interaction between Sam37 and the POTRA domain is not essential for the assembly of the SAM complex. The majority of Sam50 L6 and β-barrel interactions are between highly conserved residues, or semi-conserved residues. Additionally, on the backside of the Sam50 β-barrel (opposite the β1–β16 interface), there are two vertical lines of conserved residues (one formed with residues on β11, β10, and β8 and the other formed by residues on β10–β7). Of these highly conserved residues, two histidine sidechains (H272 on β8, H236 on β7) face the membrane and could be important for interacting with other membrane proteins, such as Mdm10[37]. There are also three highly conserved residues on the intermembrane space loop between β8 and β9 of Sam50.

**Comparison of Sam50 and BamA.** Superposition of Sam50 with BamA from *Neisseria gonorrhea* (PDB:4K3B)[20] shows that both proteins use a 16-stranded β-barrel to span the membrane, with one (Sam50) or five (BamA) POTRA domains attached at the N-terminus (Fig. 6a). The β-barrels superpose with an RMSD of 2.047 Å, with a conserved tilt angle (shear number) of the strands in the membrane. In Sam50, the POTRA domain is positioned away from the barrel lumen, potentially allowing substrate access in this conformation. In *N. gonorrhea* BamA, the POTRA domains are positioned to occlude access. Such differences have been observed in a variety of bacterial Omp85 structures. The Sam50 β-barrel is only partially closed, and PyMOL analysis does not identify any interactions at the lateral gate (Fig. 6a,

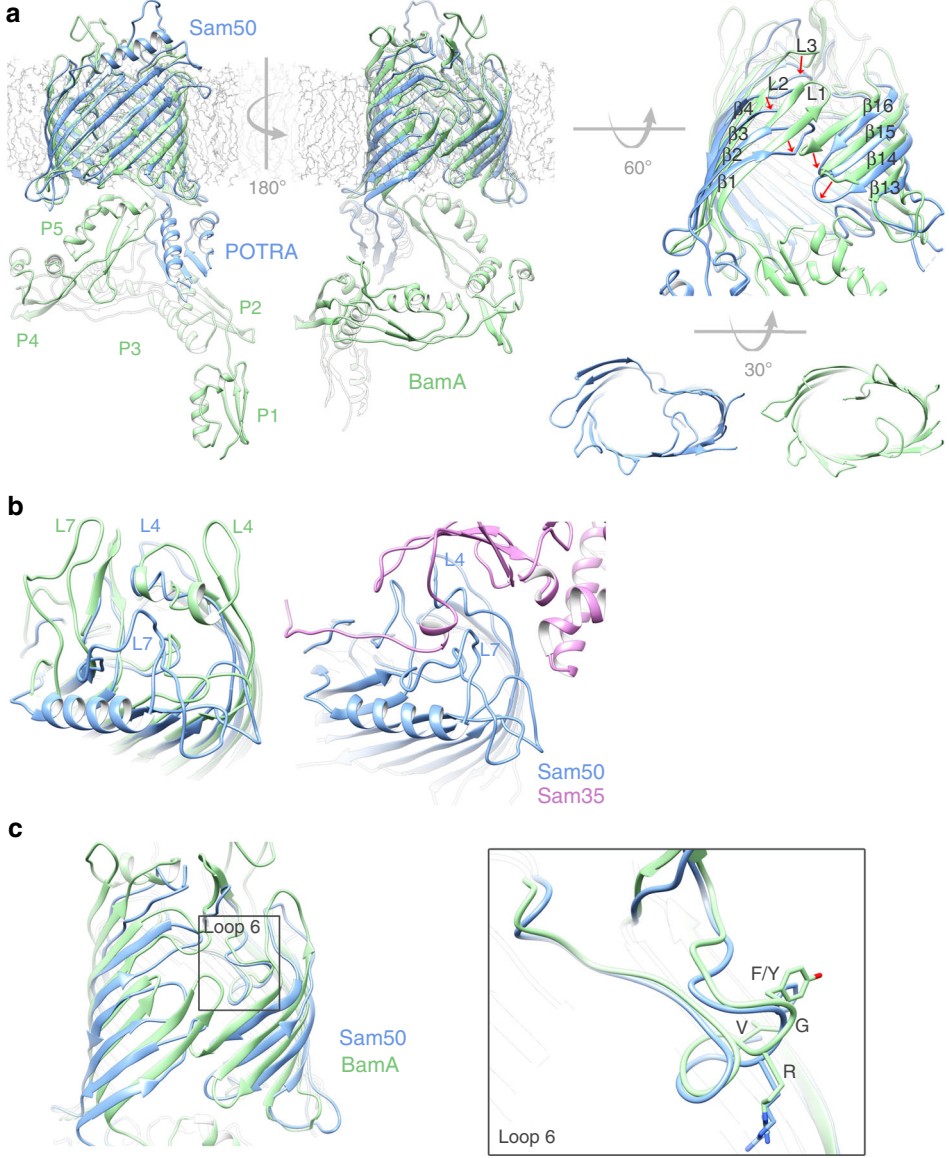

**Fig. 6 Comparison of Sam50 to BamA. a** Superposition of Sam50 (blue) and BamA (green PDB:4K3B) Right: close-up of the Sam50 lateral gate in detergent GDN compared to BamA and a bottom view of the barrel. The displacement of loops between the two structures is shown by red arrows. **b** Differences between Sam50 and BamA in outward facing loops L4 and L7 in the context of Sam35. **c** Superposition of loop 6 between the two structures. The conserved (V/I)RG(F/Y) motif is labeled. See also Supplementary Figs. 8, 9, 12, Supplementary Table 4, and Supplementary Movie 1.

Supplementary Fig. 8 and Supplementary Table 4). In contrast, the *N. gonorrhea* BamA β-barrel is closed by 4 hydrogen bonds; two between strands β1 and β16, one between loop 1 and β16, and one between β1 and a residue after the C-terminal glycine kink in β16 (Fig. 6a, Supplementary Fig. 8 and Supplementary Table 4). Strands β1 and β16 form the lateral gate that opens and closes to accommodate substrate as it folds. An important feature of Omp85 proteins is that strand β16 contains a conserved glycine in the middle that facilitates kinking of that strand so that it can fold into the lumen of the barrel. The kink is required for lateral gating in BamA[38], and this glycine is also conserved in Sam50 (Supplementary Fig. 12). In Sam50, we observe a similar kinking of strand β16 that likely facilitates lateral gate opening.

In Sam50, cytoplasmic loops L1, L2, and L3, fold in toward the barrel lumen to create the irregularly shaped barrel, whereas the BamA barrel is more elliptical (Fig. 6a). A number of other differences are observed for the outward facing loops. In BamA, loops L4 and L7 extend from the membrane surface and fold over

the top of the barrel, forming a "capping dome" that prevents substrate efflux. In Sam50, L4 and L7 adopt more open conformations and interact with the N-terminal region of Sam35; nonetheless these interactions also close the barrel lumen and prevent substrate efflux (Fig. 6b). Interestingly, loop 7 forms a 4-turn α-helix which sits parallel to the plane of the membrane. This has not been observed in BamA proteins and its function is unclear at present, although the position suggests that it may interact with the outer leaflet of the membrane.

In all Omp85 proteins, there is a conserved (V/I)RG(F/Y) motif on loop L6, where mutations to alanine abolish activity[19,39]. The conformation of the (V/I)RG(F/Y) motif is virtually identical for Sam50 and BamA, illustrating the strong structural and sequence conservation between bacteria and mitochondria (Fig. 6c). However, L6 is attached to the β-barrel of BamA very differently from Sam50.

In BamA, L6 interacts with strands β12 and β13 in the barrel lumen, using highly conserved residues (aspartate and glutamate,

respectively) to form salt bridges with a highly conserved arginine on L6. In fact, the L6 interaction with β13 is seen in all bacterial Omp85 structures solved to date. In contrast, the highly conserved L6 arginine of Sam50 forms a hydrogen bond with semi-conserved asparagine on β12 and a salt bridge with a highly conserved glutamate on β15. Sam50 L6 does not interact with β13 in our structure. The Sam50 semi-conserved asparagine on L6 forms two hydrogen bonds with β16; one with the C-terminal residue (L512) and another with S510, which is located after the glycine kink in β16. These two interactions stabilize the kinked conformation of β16. Sam50 and BamA both contain a backbone interaction between L6 and L8, which likely serves to stabilize the L6 conformation.

Although the functions of L6 are not completely clear, it seems to play a role in β-barrel stability in BamA proteins as they open and close at the lateral gate[20], while in Sam50 it plays an additional more active role, participating in substrate transfer to the lateral gate[28]. The interaction observed between β16 and L6 may help guide substrates to the lateral gate, explaining how substrates can interact with both L6 and β16 in mitochondria.

**Opening of the Sam50 lateral gate observed in SAM dimers**. 3D reconstructions for the SAM complex in GDN showed a mixture of monomers and three predominant dimer conformations (Supplementary Fig. 7). Among the dimer conformations, substantial differences were observed for the Sam50 β-barrel. Before comparing these conformations, it is important to note that detergents have the potential to greatly distort membrane protein structures, while lipid environments such as nanodiscs appear to preserve a native or native-like conformation (see for example refs. [26,27]). With these considerations in mind, we note that in dimer 1 as previously discussed, the Sam50 β-barrel is partially closed. However, in dimers 2 and 3, the β-barrel is open, allowing for association of strands β1 in an antiparallel arrangement, extending the β-sheet across two Sam50 β-barrels (Supplementary Fig. 7A). While all three dimers are clearly non-physiological, they illustrate how strand β1 of Sam50 could associate with a precursor protein[28]. A morph of Sam50 structures from dimers 1 and 3 shows how strands β1–β4 undergo an approximate 45° rotation, opening the β-barrel between strands β1 and β16 to create space for precursor protein to bind. In this morph between our observed conformations, strands β5–β16 remain mostly fixed, as does loop L6 and the conserved (V/I)RG(F/Y) motif (Supplementary Movie 1). Additional structural changes are likely to occur when substrate is present. The large conformational change of strands β1–β4 is unlike that in BamA, where the whole β1–β4 region of the barrel moves vertically and laterally. In BamA structures, the POTRA domain can move together with the base of strand β1, blocking access to the β-barrel lumen and also reducing the barrel diameter on the periplasmic side. In Sam50, the barrel diameter by the POTRA domain stays constant while the upper ¾ region of strands β1–β4 can flip outward. The POTRA domain is almost static compared to BamA. The closed conformation of the barrel in BamA is somewhat similar to the open conformation in Sam50 if we disregard the position of the POTRA domain. The POTRA domain in Sam50 never obstructs the lumen of the barrel as in BamA, being positioned more like in the open conformation of BamA.

## Discussion

In mitochondria and bacteria, precursor proteins are recognized by the SAM or BAM folding machinery through a sorting signal in the final β-strand, called the β-signal in mitochondria[3] and the C-terminal signature sequence in bacteria[40]. These signature motifs are related but not identical; for example, bacterial motifs almost always end with a C-terminal phenylalanine or tryptophan and do not tolerate extensions beyond this, while mitochondrial motifs place an additional residue after the final hydrophobic position. However, sufficient similarity between the two systems exists to allow mitochondrial VDAC expression in bacterial outer membranes[41] and bacterial OMP expression in mitochondrial outer membranes[42]. BAM and SAM are clearly similar enough to recognize one another's substrates. In both systems, the β-signal directs the precursor protein to the folding machinery and is thought to make the first interactions with it by binding at the lateral gate of Sam50 or BamA. Disulfide crosslinking experiments on SAM[28] and BAM[21] show that the β-signal of the precursor protein inserts into the lateral gate by binding to strand β1 on Sam50 or BamA, displacing the native Sam50 or BamA β-signal (strand β16) to form a stable interface between folding machinery and precursor protein. Precursor proteins also interact with strand β16 on the other side of the lateral gate, but these interactions are more flexible, suggesting that N-terminal portions of the precursor protein are added at this interface while the C-terminal precursor strand remains stably bound to β1 of Sam50 or BamA. Experiments on SAM show that Sam50 L6 is required for β-signal binding and for insertion of subsequent β-hairpins[28]. Our structures illustrate how Sam50 can open sufficiently at the lateral gate to accommodate precursor binding at strand β1, positioning L6 at the lateral gate through its interaction(s) with strand β16. Future experiments will explore interactions between Sam50 strands β1, β16, and L6 with a true β-signal peptide.

Sam35 specifically recognizes β-signal peptides independently of Sam50[3], but from a structural standpoint it remains unclear how this interaction occurs. Our analysis of the semi-conserved residues on Sam35 identified a groove with increased conservation which could be a potential binding pocket. However, this region is on the cytosolic side of the membrane which poses a topological conundrum, because SAM complex substrates reside in the intermembrane space prior to their folding and insertion into the outer membrane. Additionally, the top of the Sam50 β-barrel is occluded by the cytoplasmic loops of Sam50 and the N-terminus of Sam35, so it is less likely that substrates will pass all the way through the barrel to interact with this semi-conserved groove on Sam35. While the N-terminal residues of Sam35 located across the top of Sam50 are not well conserved, they would be in a closer position to interact with substrate inside the Sam50 β-barrel.

Sam50 transiently associates with Mdm10 in the outer mitochondrial membrane to aid in TOM complex biogenesis[43]. In *S. cerevisiae* Mdm10, two conserved aromatic residues (Y73 and Y75) are important for association with the SAM complex, and mutation of these residues disrupts the Sam50–Mdm10 interaction as well as Tom40 and Tom22 biogenesis[37]. Homology models of Mdm10 predict Y73 and Y75 are located on strand β3 within the mitochondrial outer membrane[37,44]. Analysis of Sam50 sequence conservation mapped to our structure identifies two stretches of highly conserved residues on strands β7–β11, which are opposite of the lateral gate (Supplementary Fig. 12). These highly conserved residues on Sam50, particularly those on β7 and β8 (H236 and H272, respectively), are in a reasonable position to interact with the conserved Mdm10 residues important for Sam50–Mdm10 interaction. Experimental evaluation of the involvement of these highly conserved Sam50 residues in the interaction with Mdm10 will be required. It is not currently understood how the Sam50–Mdm10 interaction contributes to TOM complex biogenesis.

## Methods

**Cloning of expression vectors**. *Myceliophthora thermophila* (recently renamed *Thermothelomyces thermophilus*; we will continue to use *M. thermophila* here)

SAM complex subunit coding sequences were codon optimized for expression in *S. cerevisiae* and obtained from GenScript. Sam50, Sam35, and Sam37 genes were cloned into pBEVY expression vectors after the GAL1 promoter using In-Fusion Cloning (Takara Bio USA, Inc) or golden gate cloning. The Sam37 construct included a N-terminal Twin-Strep-tag® (IBA GmbH) and a glycine-glycine linker. Each construct was verified by sequence analysis (Macrogen USA). A complete list of primers used in this study are in Supplementary Table 5.

**Yeast transformation and growth.** The three SAM complex pBEVY vectors were co-transformed into *Saccharomyces cerevisiae* strain W303.1B (MATa {leu2-3,112 trpl-1 canl-100 ura3-1 ade2-1 his3-11,15}) using the lithium acetate/single-strand carrier DNA/PEG method[45–47]. Transformed cells were plated on a selection agar (6.9 g/L yeast nitrogen base without amino acids, 0.62 g/L Clontech -Leu/-Trp/-Ura dropout supplement, 20 g/L bacto agar, 2% D-(+)-glucose) and incubated for 72 h at 30 °C.

One colony was used to inoculate 10 mL of selection media (6.9 g/L Yeast Nitrogen Base without amino acids, 0.62 g/L Clontech -Leu/-Trp/-Ura dropout supplement, 2% D-(+)-glucose) and incubated overnight at 30 °C and 220 rpm. The following day, 500 mL selection media in a 1 L glass flask was inoculated with the 10 mL starter culture and incubated overnight at 30 °C and 220 rpm. Twelve 2 L glass flasks containing 1 L YPG media (10 g/L yeast extract, 20 g/L peptone, 30 mL glycerol, 0.1% glucose) were each inoculated with starter culture to reach a start OD$_{600nm}$ of 0.15 and incubated for 16–18 h at 30 °C and 220 rpm. Expression was induced with 0.4% D-galactose (final concentration), and cultures were allowed to grow for 4 h at 30 °C and 220 rpm. Cells were harvested and washed with cold sterile ultra-pure water before storing at −80 °C.

**Mitochondrial isolation.** Thawed cell pellet was resuspended in breaking buffer (650 mM sorbitol, 100 mM Tris-HCl, pH 8.0, 5 mM EDTA, pH 8.0, 5 mM amino hexanoic acid, 5 mM benzamidine, 0.2% BSA) and stirred for 30 min at 4 °C. Once resuspended, 4 mL of 200 mM PMSF (PMSF in 100% ethanol) was added. Resuspended cells were passed through a Dyno-Mill Multi Lab (WAB) bead mill (0.5–0.75 μm glass beads) at a flow rate of 35 mL/min and the chamber temperature was maintained below 10 °C[48]. Disrupted cells were collected on ice and 4 mL 200 mM PMSF was added.

Cell debris was removed by two centrifugation steps, transferring the supernatant to fresh tubes after the first (3470 × g, 30 min each, 4 °C). Mitochondrial membrane sample was isolated by centrifugation (24,360 × g, 50 min, 4 °C). The pellet was resuspended in wash buffer (650 mM sorbitol, 100 mM Tris-HCl, pH 7.5, 5 mM amino hexanoic acid, 5 mM benzamidine) using a Dounce homogenizer on ice. Sample was centrifuged again (24,360 × g, 50 min, 4 °C). Pellet was resuspended by homogenization in Tris-buffered glycerol (TBG) mitochondrial storage buffer (100 mM Tris-HCl, pH 8.0, 10% glycerol), and centrifuged once more (24,360 × g, 50 min, 4 °C). Pellet was resuspended by homogenization in TBG storage buffer and separated into three aliquots. Protein concentration of the mitochondrial membrane sample was determined using Pierce BCA Protein Assay Kit (Thermo Fischer Scientific). Mitochondrial membrane aliquots were flash frozen in liquid nitrogen and stored at −80 °C.

**SAM complex nanodisc incorporation.** One aliquot of mitochondrial membrane was thawed, and the protein concentration was adjusted to 10 mg/mL with TBG mitochondrial storage buffer. Sodium chloride was added to a final concentration of 150 mM, and Roche cOmplete Protease Inhibitor Cocktail tablets were added. Sample was mixed with stir bar at 4 °C until the protease inhibitor tablets dissolved. Membrane was solubilized by addition of 2% final concentration of LMNG (Anatrace) and stirring for 1.5 h at 4 °C. Solubilized material was isolated by ultracentrifugation (208,000 × g, 45 min, 4 °C). The supernatant was filtered with a 0.22 μm SteriFlip vacuum filter (Millipore) and used immediately.

Filtered supernatant was added to Strep-Tactin Sepharose (IBA GmbH) resin and rocked for 4 h at 4 °C. Following incubation, sample was transferred to a gravity flow column and washed with four column volumes of wash buffer (100 mM Tris-HCl, pH 8.0, 150 mM NaCl, 1 mM EDTA, pH 8.0, 0.02% LMNG). Protein was eluted with elution buffer (100 mM Tris-HCl, pH 8.0, 150 mM NaCl, 1 mM EDTA, pH 8.0, 2.5 mM Desthiobiotin, 0.02% LMNG). Fractions were analyzed by SDS PAGE, and those containing SAM complex were pooled and concentrated using 100 kDa molecular weight cut off Amicon Ultra Centrifugal Filter Unit (Millipore).

Concentrated sample was injected onto HiLoad 16/600 Superose 6 prep grade column (GE Healthcare) at a flow rate of 0.11 mL/min, in size-exclusion buffer (20 mM HEPES, pH 8.0, 150 mM NaCl, 0.02% LMNG). Fractions were collected and analyzed by SDS-PAGE. MSP1E3D1 plasmid (Addgene) was expressed in *E. coli* (BL21 (DE3) strain, NEB) and purified using Nickel-NTA affinity chromatography as described[49]. The histidine tag was removed via TEV cleavage, MSP1E3D1 was concentrated and flash frozen in liquid nitrogen before storing at −80 °C. Purified SAM complex in LMNG, MSP1E3D1, and cholate solubilized DOPE (Avanti Polar Lipids, Inc.) and DOPC (Anatrace) were mixed in a molar ratio of 1 SAM: 2 MSP1E3D1: 200 DOPE: 200 DOPC. Additional cholate was added to obtain final concentration of

20 mM. Mixture was incubated for 1 h on ice, then 0.1 g Bio-Beads SM2 (BioRad) were added per milliliter of incorporation sample to facilitate nanodisc formation. Sample with Bio-Beads was rocked at 4 °C overnight. An additional 0.1 g/mL Bio-Beads SM2 were added the following morning and sample was rocked for 1 h at 4 °C. Bio-Beads were removed from sample with low speed centrifugation. Insoluble protein was separated by ultracentrifugation (208,000×g, 45 min, 4 °C).

Soluble fraction was added to Strep-Tactin Sepharose (IBA GmbH) resin and rocked for 4 h at 4 °C. Sample was transferred to a gravity flow column and washed with wash buffer (100 mM Tris-HCl, pH 8.0, 150 mM NaCl, 1 mM EDTA, pH 8.0). SAM complex in nanodiscs was eluted with elution buffer (100 mM Tris-HCl, pH 8.0, 150 mM NaCl, 1 mM EDTA, pH 8.0, 2.5 mM Desthiobiotin). Fractions were analyzed with SDS-PAGE and those containing SAM complex in nanodiscs were pooled and concentrated to 1 mL using 100 kDa molecular weight cut off Amicon Ultra Centrifugal Filter Unit (Millipore).

Sample was injected onto Superose 6 10/300 column (GE Healthcare) at a flow rate of 0.20 mL/min in size-exclusion buffer (20 mM HEPES, pH 8.0, 150 mM NaCl). Fractions were collected and analyzed by SDS-PAGE. The peak fraction containing SAM complex in nanodisc was concentrated to ~1 mg/mL for cryo-electron microscopy.

**SAM complex purification in detergent.** SAM complex was purified as described above, but detergent was exchanged to 0.02% GDN (Anatrace) during size-exclusion chromatography. Briefly, isolated mitochondria were solubilized in 2% LMNG for 1.5 h at 4 °C. Filtered soluble fraction was bound to Strep-Tactin Sepharose (IBA GmbH) resin for 4 h, then SAM complex was eluted with Desthiobiotin elution buffer. Concentrated sample was injected onto HiLoad 16/600 Superose 6 prep grade column (GE Healthcare) at a flow rate of 0.11 mL/min, in size-exclusion buffer (20 mM HEPES, pH 8.0, 150 mM NaCl, 0.02% GDN). Fractions were collected and analyzed by SDS-PAGE. Peak fractions containing SAM complex were pooled and concentrated to ~5 mg/mL for cryo-electron microscopy.

**Pulldown assay for the Sam35 N-terminal truncation mutant.** Sam35 (Δ1-45) truncation was generated with Q5 Site-Directed Mutagenesis (NEB). Ternary SAM complex containing Sam37 with N-terminal Twin-Strep tag and Sam35 + Sam37 complex containing C-terminal Twin-Strep tag were transformed and expressed in *S. cerevisiae*, as described above for the full-length complex.

Strep affinity pull down assays conducted in parallel, starting with 90 mg of mitochondria for each coexpression. Mitochondrial protein concentrations were adjusted to 10 mg/mL using TBG mitochondrial storage buffer. Sodium chloride was added to a final concentration of 150 mM, and one Roche cOmplete Protease Inhibitor Cocktail tablet was added to each solubilization sample. Samples were mixed with stir bar at 4 °C until the protease inhibitor tablet dissolved before adding LMNG (Anatrace) to a final concentration of 2% and stirring for 1.5 h at 4 °C. Soluble fraction separated by ultracentrifugation (208,000×g, 45 min, 4 °C) then filtered with a 0.22 μm SteriFlip vacuum filter (Millipore). Filtered supernatant was added to Strep-Tactin Sepharose (IBA GmbH) resin and rocked for 4 h at 4 °C. Following incubation, sample was transferred to a gravity flow column and washed with ten column volumes of wash buffer (100 mM Tris-HCl, pH 8.0, 150 mM NaCl, 1 mM EDTA, pH 8.0, 0.02% LMNG). Protein was eluted using elution buffer (100 mM Tris-HCl, pH 8.0, 150 mM NaCl, 1 mM EDTA, pH 8.0, 2.5 mM Desthiobiotin, 0.02% LMNG). Fractions were analyzed by SDS PAGE.

**CryoEM sample preparation and data collection.** An aliquot of 3 μL of SAM complex sample was applied to freshly glow-discharged holey carbon grid (Quantifoil R1.2/1.3, copper, 300 mesh). The grids were blotted for 6 s and plunge-frozen in liquid nitrogen-cooled liquid ethane using an FEI Vitrobot Mark IV plunger. CryoEM data were collected on a Titan Krios G3 electron microscope (Thermo-Fisher) operated at 300 kV and equipped with a Gatan Quantum LS imaging energy filter with the slit width set at 20 eV. Micrographs were acquired on a K2 Summit direct electron detection camera at the nominal magnification of ×130,000 (calibrated pixel size of 1.06 Å on the sample level) using the Leginon automation software package[50]. The dose rate on the camera was set to 8 e⁻/pixel/s. The total exposure time of each micrograph was 10 s fractionated into 50 frames with 0.2 s exposure time for each frame. Detailed data collection parameters are listed in Table 1.

**Image processing.** All frames of each dose-fractionated micrograph were aligned for correction of beam-induced drift using MotionCor2[51]. Two average images were generated from motion correction for each micrograph: one with dose weighting and the other one without. The average images without dose weighting were used for defocus determination using CTFFIND4[52]. Quality of the micrographs was evaluated using the results from CTFFIND4. The micrographs with poor resolution (worse than 4.5 Å) or too large (>3.0 μm) or too small (<0.8 μm) underfocus values were removed. The single particle data analysis was performed following the standard procedures in RELION3[53] and cryoSPARC[54] with few modifications as summarized in the following and in Supplementary Figs. 3C and 5.

**SAM complex in lipid nanodiscs**. The procedures are summarized in Supplementary Fig. 3C. No symmetry was applied in the data processing of the SAM complex in lipid nanodiscs. A total of 14,032 micrographs were acquired and 11,347 micrographs were selected for data processing using the results from CTFFIND4. Initially, a set of 500 micrographs were used for particle picking by Gautomatch (https://www.mrc-lmb.cam.ac.uk/kzhang/Gautomatch/). No external templates were supplied and Gautomatch generated templates automatically. The particles were processed using cryoSPARC2 for 2D classification and ab-initio reconstruction to generate starting models. The projections of the best starting model was then used as templates for particle picking by Gautomatch using all micrographs. A total of 3,951,406 particles were picked and extracted in 96 × 96 pixels with 2× binning (pixel size 2.12 Å). The particles were processed using RELION3 for 2D and 3D classifications. The best particles were selected iteratively by selecting the 2D class averages and 3D reconstructions that had interpretable structural features. 446,747 particles were selected using the above procedures and subjected to 3D classification with only 1 class to align particles to the center. The result was used to re-center the particles as well as to remove overlapping particles (deduplication), leading to 445,569 particles being re-extracted in 192 × 192 pixels without binning (pixel size 1.06 Å). After further 2D and 3D classifications, 205,131 particles were selected for 3D autorefinement to obtain a reconstruction at 4.2 Å resolution. The result was then used for particle polishing that included all frames. A 2D classification was always carried out after each run of particle polishing to remove particles containing bad pixels from the camera in the processing of all the data sets reported here. The resolution of 3D autorefinement was improved to 4.0 Å with 190,078 particles after particle polishing. Ctf refinement was then performed to refine per-particle defocus values but no resolution improvement was noticed at this step. An additional run of particle polishing was carried out using the first 30 frames and 179,509 "shiny" particles were generated for the final 3D autorefinement using RELION3 or cryoSPARC2. The refinement using RELION3 reported 3.9 Å resolution and the non-uniform refinement using cryoSPARC2 reported 3.4 Å resolution. A soft mask was used in the RELION3 3D autorefinements. The cryoSPARC2 non-uniform refinement also employed an auto-generated soft mask. In our case, the cryoSPARC2 non-uniform refinement generated better results as reported by FSC and the quality of the density maps, therefore the 3D reconstructions from cryoSPARC2 were used for structural interpretation and atomic modeling.

**SAM complex in detergent**. The procedures are summarized in Supplementary Fig. 5. A total of 14,329 micrographs were acquired and 10,831 micrographs were selected using the results from CTFFIND4. Similar to the above procedures, a small subset of micrographs were used for particle picking without a template. The particles were processed for 2D classification using RELION3. Representative 2D class averages were selected to serve as templates for a new iteration of particle picking by Gautomatch using all the micrographs. A total of 5,842,605 particles were picked and extracted in 104 × 104 pixels with 2x binning (pixel size 2.12 Å). The particles were processed for 2D classifications using RELION3. The best particles were selected iteratively from the results of 2D classification. Comparison of the 2D class averages suggested that there are two different forms of complexes in the sample: "monomeric" and "dimeric" complexes (Supplementary Fig. 5). These two different forms of particles were separated (514,800 monomer particles and 829,206 dimer particles) for the following processing procedures.

The processing of monomer particles is summarized in Supplementary Fig. 5. After re-centering and deduplication, 511,249 monomer particles were re-extracted in 208 × 208 pixels without binning (pixel size 1.06 Å) and subjected to 2D classification using RELION3, from which 457,279 particles were selected for further processing using cryoSPARC2. No symmetry was applied in the data processing of the monomer particles. After particle selection from 2D classification and 3D hetero refinement, 138,575 particles were used in non-uniform refinement to achieve a final reconstruction at 3.7 Å resolution.

The processing of dimer particles is summarized in Supplementary Fig. 5. After re-centering and deduplication, 821,025 dimer particles were re-extracted in 208 × 208 pixels without binning (pixel size 1.06 Å) and subjected to 2D and 3D classifications using RELION3. A starting model of the dimer was generated using the ab-initio reconstruction in cryoSPARC2 to serve as the reference map for the 3D classification in RELION3. The C2 symmetry was applied in all 3D reconstructions during the data processing of the dimer particles. A total of 5 classes of 3D reconstructions were generated by 3D classification. 3 of the 5 classes show interpretable structural details and their conformations are notably different, therefore the particles in these 3 classes were separately sent to cryoSPARC2 for further hetero refinement and non-uniform refinement to obtain final 3D reconstructions. It is worth noting that the particles of "dimer 1" were processed using RELION3 for 3D autorefinement, Ctf refinement, and particle polishing before the cryoSPARC2 processing. The resolutions of the 3D reconstructions of these dimers are 3.2 Å ("dimer 1" with 117,339 particles), 3.6 Å ("dimer 2" with 60,472 particles), and 3.9 Å ("dimer 3" with 122,361 particles), respectively.

The interfaces between two monomers in the dimers are relatively small and suspectable to cause flexibilities in the dimers as evidenced by poor densities in some regions of the 3D reconstructions (Supplementary Fig. 6D). Therefore, the reconstruction of "dimer 1", which is at the highest resolution among three different dimers, was further improved by dividing the dimer into two independent monomers for refinement using symmetry expansion (Supplementary Fig. 6D). Briefly, 174,217 polished particles of "dimer 1" were processed using RELION3 for Ctf refinement and 3D autorefinement with the C2 symmetry. Each particle was then expanded into two particles using the program *relion_particle_symmetry_expand* to generate a total of 335,670 particles. No symmetry was applied in the afterward procedures. A soft mask, that included one monomer and the detergent micelle, was carefully constructed following the instructions[55] for density subtraction. Another soft mask that only included the remaining monomer after density subtraction was also constructed for the following 3D refinements. During RELION3 3D autorefinement, local search was forced by setting both parameters "initial angular sampling" and "local searches from auto-sampling" to 1.8°. A 3D reconstruction of the monomer was obtained at 3.3 Å resolution after the first run of 3D autorefinement using the density-subtracted particles. Then the particles were subjected to particle polishing for a second time using the first 30 frames. It is worth noting that the particles after this step of particle polishing were intact dimers without density subtraction. The soft mask and local search were used in the 3D refinement of the monomer with the polished particles of dimer to ensure the cross interferences between two monomers within each particle was minimal. Finally, the polished "shiny" particles were sent to cryoSPARC2 for a final 3D refinement with the features of local refinement and non-uniform refinement turned on to achieve the 3D reconstruction of the monomer at 3.0 Å resolution. The resolution and quality of the reconstruction were both improved by symmetry expansion as shown in Supplementary Fig. 6D.

**Model building and refinement**. A model was manually built into the final density map of the SAM complex in lipid nanodiscs using COOT[56]. The protein sequences of *M. thermophila* Sam35, Sam37 and Sam50 (Uniprot: G2QAT9, G2Q6R7, and G2QFF9 respectively) were used to build models from scratch. Secondary structure predictions were performed using I-TASSER[57] and Phyre2[58]. At the level of 3.4 Å resolution, the density was sufficient to allow ab initio tracing of a majority of the folds of all three subunits. This model was later used as an initial reference for model building into the density maps of dimeric SAM complexes in detergent. In the highest resolution (3 Å) map of the monomeric SAM complex derived from dimer 1, a majority of the protein sidechains were well resolved, allowing model building without ambiguity. All the models were refined using both Rosetta[59] and the real-space refinement in PHENIX[60]. The statistics are summarized in Table 1.

**Sequence alignment**. Sequences for SAM complex subunits from different species were obtained from UniProtKB[61]. Multiple sequence alignment of each SAM complex subunit was completed using T-Coffee Expresso[62–66]. Single residue manual adjustments to the sequence alignment were completed in Jalview[67]. The final alignment with sequence similarities highlighted was produced with ESPript 3.0, with sequence similarities depiction parameters set to %Equivalent, global score of 0.6, and flashy color scheme output[68]. Lastly, the high-resolution structure from SAM complex in detergent was used to assign secondary structure to the alignment.

**Interaction analyses**. The interaction analyses for Sam50/Sam35, Sam35/Sam37, and Sam50/37 were completed using QT Pisa[69] and PyMOL (Version 2.3 Schrödinger, LLC), and figures were made using Chimera[70,71].

**Reporting summary**. Further information on research design is available in the Nature Research Reporting Summary linked to this article.

## Data availability
Data supporting the findings of this manuscript are available from the corresponding authors upon reasonable request. A reporting summary for this Article is available as a Supplementary Information file. Atomic coordinates and structure factors for SAM complex structures have been deposited in the EMDB under accession codes EMD-21913, EMD-21914, EMD-21915, EMD-21916, EMD-21917, and EMD-21918 and wwPDB under accession codes PDB 6WUH [https://doi.org/10.2210/pdb6WUH/pdb] PDB 6WUJ [https://doi.org/10.2210/pdb6WUJ/pdb] PDB 6WUL [https://doi.org/10.2210/pdb6WUL/pdb] PDB 6WUM [https://doi.org/10.2210/pdb6WUM/pdb] PDB 6WUN [https://doi.org/10.2210/pdb6WUN/pdb] PDB 6WUT [https://doi.org/10.2210/pdb6WUT/pdb] *Myceliophthora thermophila* SAM complex sequences are available from uniprot.org under UniProt IDs G2QAT9 (Sam35), G2Q6R7 (Sam37), and G2QFF9 (Sam50). The source data underlying Supplementary Figs. 2 and 13 are provided as a Source Data file. Source data are provided with this paper.

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

## Acknowledgements

This work utilized the NIH Multi-Institute Cryo-EM Facility (MICEF) and the computational resources of the NIH HPC Biowulf cluster (http://hpc.nih.gov). We thank Huaibin Wang and Haifeng He for technical support on the NIH MICEF Titan Krios Electron Microscope. K.A.D., S.E.R., I.B., and S.K.B. are supported by the Intramural Research Program of the NIH, NIDDK. X.N., X.T., and J.J. are supported by the Intramural Research Program of the NIH, NHLBI. S.R. was also supported by a Sir Henry Wellcome Postdoctoral Fellowship. M.S.K. and E.R.S.K. are funded by program grant MC_UU_00015/1 of the Medical Research Council. We thank Travis Barnard for help with the design and cloning of the pBEVY vectors and Bridgette Beach for helping to clone the Sam50 POTRA truncation construct.

## Author contributions

K.A.D., S.E.R., E.R.S.K., and S.K.B. designed the study. K.A.D., S.E.R., and M.S.K. cloned, expressed and purified SAM complexes. X.N. and J.J. collected and processed cryoEM data. X.N. and X.T. built the initial models. X.N., I.B., and K.A.D. refined the final models. All authors analyzed the data and wrote the manuscript.

## Competing interests

The authors declare no competing interests.
