## [Peer Review File · Nature Communications]

Reviewers' comments:

Reviewer #1 (Remarks to the Author):

The SAM complex (known also as the TOB complex) functions in the assembly of newly synthesized β -barrel proteins into the mitochondrial outer membrane. The central component of this complex, Sam50 (named also Tob55) is a member of the extended Omp85 family and homologous to the bacterial BamA. Sam50 is a β -barrel protein itself and forms the SAM complex with two peripheral subunits, Sam37 (Mas37) and Sam35 (Tob38), that are placed on the cytosolic face of the membrane.

Several biochemical studies shed light on the functions of the individual Sam subunits. However, structural information on the complex at atomic resolution is missing. In previous reports Buchanan and her co-workers modeled Sam50 based on its similarity to BamA.

In the present contribution, Diederichs et al. provide a near atomic structure of the SAM complex. In this structure, they observed that Sam35, on the one hand, caps the lumen of Sam50 and, on the other hand, interacts intensively with Sam37. Their structure supports a lateral opening of the barrel to explain how the complex accommodate substrate proteins. Based on the structure, the authors suggest a model for the interaction of the SAM complex with the TOM complex.

Major points:

1. The manuscript will benefit from some biochemical assays to support the structural model.

(i) amino acids that were suggested to contribute to the interactions of Sam35 with Sam50 can be mutated and the outcome regarding the structure (cryo-EM), assembly (BN-PAGE), and the function of the SAM complex (growth assays) should be monitored.

(ii) residues of Sam37 that were suggested to interact with the POTRA domain of Sam50 should be mutated. Then, the impact on the structure of the complex, the association of Sam37 with the other two subunits (BN-PAGE), as well as the function of Mas37 (complementation of yeast strains lacking MSA37) should be investigated.

(iii) site-specific cross-linking assays can support the involvement of specific residues in protein-protein interactions.

In its current form, this contribution does not provide much new insight into the mechanism of membrane assembly of β -barrel proteins. The lateral gate was already suggested based on the modelling of Sam50 on the structure of BamA (ref. 19) and biochemical assays (ref. 23).

2. The modelling of the SAM-TOM supercomplex is based on very limited data and neglects the presence of the receptors Tom20 and Tom70 in native membranes. Thus, the statement that the supercomplex 'has no major clashes between members of the SAM and TOM complexes' is not accurate because some Tom subunits were neglected. I suggest that the authors will either support their model with more experimental data (biochemical or structural) or will remove this part.

Minor points:

1. Lines 82-3: This statement is not completely correct since newly synthesized Sam50 molecules are probably also substrates of pre-existing SAM complexes.

2. An SDS-PAGE of the purified SAM complex should be shown to evaluate the purity of the complex. In addition, the size exclusion behavior of the purified complex should be presented.

3. The authors failed to cite the article of Klein et al. (JCB 2012) who presented a low-resolution structure of the TOB complex from *N. crassa* and reported a stoichiometry of 1:1:1 among the

complex components.

4. The comparison of the structure of Sam50 to that of BamA takes a major part of this contribution and does not necessarily add mechanistic insights. I would suggest shortening this part.

Reviewer #2 (Remarks to the Author):

Mitochondrial β -barrel proteins play important roles in the outer membrane of mitochondria. After being synthesized in the cytosol, these proteins are imported across the mitochondrial outer membrane as unfolded precursor proteins, and then interact with the Sorting and Assembly Machinery (SAM) complex, which mediates the folding and membrane insertion of β -barrel proteins. The SMA complex contains one copy of three proteins: Sam50, Sam35 and Sam37. Sam50 is a 16-stranded β -barrel protein, and the other two proteins are located on the cytoplasmic side of the mitochondrial outer membrane. In this manuscript "Structural insight into mitochondrial β -barrel outer membrane protein biogenesis", the authors determined the cryo-EM structures of the SAM complex in lipid nanodiscs and in detergents at 3-4 Å resolutions. The SAM complex in nanodisc is monomeric, whereas it forms various dimers in the GDN detergent. Their structural analysis revealed the details of interactions between SAM components, suggested conformational dynamics of Sam50 β -barrel, and compared the structures of Sam50 and BamA, another member in the same Omp85 family. Cryo-EM work was nicely done, and the manuscript is well written. Their findings are interesting to the researchers in the related fields. However, a few important issues should be addressed before publication.

(1) It is surprising to find absolutely no biochemical data in the manuscript. SDS-PAGE and gel-filtration profiles are essential to evaluate the protein homogeneity. This is particularly relevant to the apparent discrepancy of oligomer states of the SAM complex in nanodiscs and GDN detergent. The authors should show gel-filtration profiles for the SAM complex in LMNG, GDN and nanodiscs.

(2) Are there different oligomer states for SAM in nanodiscs, LMNG, or GDN? If so, are the distributions of different oligomers similar in these samples? Did GDN particularly induce dimer formation? Have the authors performed structural analysis on other oligomers of SAM in nanodisc or on SAM in LMNG?

(3) In the section of "Opening of the Sam50 lateral gate observed in SAM complex dimers", it is interesting and reasonable to speculate the β -strand interaction between substrate protein and Sam50. However, the conformational variation between SAM structures in nanodisc and in the GDN detergent should be treated with great caution. The possible effects from particular detergents on Sam50 structure and open/close β -barrel conformations should be discussed.

(4) The modeling of TOM-SAM supercomplex is interesting, but highly speculative without the support of more structural and biochemical data. Therefore, current Fig. 6 seems to be more appropriate as a supplementary figure.

(5) For a general reader, a schematic at the beginning of the manuscript would be helpful, to illustrate the structural features in SAM, and to compare SAM and BAM. This should indicate various domains including all β -strands, loop names, and Sam37 TM and linker. This can also be done in Fig. 1, when the SAM structure is first presented.

(6) Line 160. "... the most prevalent dimer (dimer 1)". This seems a wrong statement according to Suppl. Fig. 2. Dimer 3 appears to have more particles in both 1st round 3D classification and in the final refinement step. What is the reason for the authors to focus on dimer 1? What would be the results if the same extensive image processing procedure for dimer 1 is applied on dimer 2 and

dimer 3?

(7) In several parts of the manuscript, the analysis/presentation would make more sense if the points in the text are indicated in figures.

Fig.1. Indicate aa 339-353 (related to text in line 140) and the presumable position of TM helices of Sam37, and POTRA domain of Sam50.

Line 228-242 has detailed discussion about interactions between conserved residues. But not a single residue was indicated in Supplementary Fig. 6, even though the levels of conservation were colored.

Line 258. Where is this conserved glycine in the structure?

Line 144. "At low resolution ..." and Line 145 "at high resolution" are perhaps better to change to "When filtered to low/high resolution...". In addition, these maps filtered to different resolutions should be shown to support the claims.

Other minor points:

Can the authors explicitly list all available structures of Omp85 members in Introduction?

Throughout the text and methods, many "Fig. S#" should be changed to "Supplementary Fig. #".

Fig. 4. The right panels do not exactly match the left panels. It would be helpful to label certain residues in the right panels to orient the readers.

Suppl. Fig. 1C. The final step is using cryoSPARC with non-uniform refinement. Can the authors show the results from (1) CryoSPARC without non-uniform refinement, and (2) Relion auto-refine? This would be interesting and informative for the users of these programs.

Suppl. Fig. 1E and 3B. Among these FSCs, only the 3rd and 5th are essential. However, these figures are missing two validation FSCs: [model refined against half map1] vs. [half map1], and [model refined against half map1] vs. [half map2]. The consistency between these two FSCs would prove no over-refinement of the model.

Suppl. Fig. 3D. What are the sigma levels for display?

Suppl. Table 3. What is the criteria/threshold for assigning a residue as "conserved"?

We thank the reviewers for their instructive comments and have addressed them all below:

Reviewers' comments:

Reviewer #1 (Remarks to the Author):

The SAM complex (known also as the TOB complex) functions in the assembly of newly synthesized β -barrel proteins into the mitochondrial outer membrane. The central component of this complex, Sam50 (named also Tob55) is a member of the extended Omp85 family and homologous to the bacterial BamA. Sam50 is a β -barrel protein itself and forms the SAM complex with two peripheral subunits, Sam37 (Mas37) and Sam35 (Tob38), that are placed on the cytosolic face of the membrane.

Several biochemical studies shed light on the functions of the individual Sam subunits. However, structural information on the complex at atomic resolution is missing. In previous reports Buchanan and her co-workers modeled Sam50 based on its similarity to BamA.

In the present contribution, Diederichs et al. provide a near atomic structure of the SAM complex. In this structure, they observed that Sam35, on the one hand, caps the lumen of Sam50 and, on the other hand, interacts intensively with Sam37. Their structure supports a lateral opening of the barrel to explain how the complex accommodate substrate proteins. Based on the structure, the authors suggest a model for the interaction of the SAM complex with the TOM complex.

Major points:

1. The manuscript will benefit from some biochemical assays to support the structural model.

(i) amino acids that were suggested to contribute to the interactions of Sam35 with Sam50 can be mutated and the outcome regarding the structure (cryo-EM), assembly (BN-PAGE), and the function of the SAM complex (growth assays) should be monitored.

We fully agree with the reviewer that biochemical assays would benefit the manuscript. However, this is a fairly large protein complex and point mutations would likely tell us nothing about the interaction interfaces. The sizes of these interaction interfaces are substantial, ranging from $\sim 950 \text{ \AA}^2$ to $\sim 1980 \text{ \AA}^2$, with 9-18 H-bonds and 0-8 salt bridges per protein molecule. Most of the H-bonding interactions are between main chain atoms, therefore mutating side chains will have no effect on these. For Sam37-Sam35 out of 9 H-bonds, 3 are strictly between main chain atoms, 4 between main chain:side

chain and 2 strictly between side chains. There are only 2 salt bridges between Sam37-Sam35. Between Sam37 and Sam50 there are no salt bridges and 8 H-bonds. 4 H-bonds strictly between main chain atoms and 4 between side chain:main chain. Between Sam50 and Sam35 there are 3 salt bridges and 15 H-bonds. 7 of the H-bonds are strictly main chain, 5 are hybrid and 3 are strictly between side chains. Mutating all the residues involved in H-bonding or salt bridges would imply construction of 58 individual mutants but taking into account the main chain interactions reduces the number to 31. One cannot be certain that mutating these residues would disrupt the surface interactions sufficiently (leading to false negatives) or have misfolded proteins, leading to false positives.

To attempt to analyze potential mutagenesis outcomes, we performed *in silico* simulations of some of these point mutations that indicated a very small drop in the binding energy of the interaction interface per mutant (maximum 1 kcal/mol); see Appendix 1 at the end of this response letter. This assumes an ideal case without additional protein dynamics. The point mutations can also be compensated by slight rearrangements of neighboring residues in the interface, something that could be predicted only by molecular dynamics simulations. Both the construction and testing of 31 mutants and extensive MD simulations of some of the mutants are standalone projects by their sheer size. So, if individual point mutants would not make much of an impact, then why not make double or triple mutants that may have a larger impact on the interface? The possible combinations of multiple mutants that can be constructed from 31 sites is very large and therefore not possible to do for this manuscript. Instead, we performed *in silico* simulations of some of the triple mutations and the maximum binding energy difference compared to wild type was 2.3 kcal/mol. For another triple mutant the maximum solvation energy difference (ΔG) compared to wild type was 3.3 kcal/mol. This does not account for rearrangement of neighboring residues, which is more likely to occur with larger structural changes. Sometimes it is not the individual interactions but rather the shape or the total surface area of interactions that counts.

Therefore, we suggest that mapping interaction interfaces by mutagenesis may not yield new information.

However, we did still want to provide mutagenesis data for Sam50/Sam35. We made a deletion mutant for Sam35 that lacks the 45 N-terminal residues that specifically interact with Sam50. We evaluated SAM complex association by Strep affinity pull down using the Twin-Strep tag on Sam37. Elution fractions of the full-length ternary SAM complex contained approximately stoichiometric amounts of each subunit. In the Sam35 N-terminal truncation ternary SAM complex, a reduced amount of Sam50 was observed in elution fractions relative to Sam35 and Sam37. In this sample, the ratio of Sam37 and Sam35 was approximately equal (Supplementary Figure 13A and B). The deletion of the Sam35 N-terminal 45 residues disrupts the majority of the interactions identified between Sam35 and Sam50, which is supported by the reduction of Sam50 present in the pulldown elution fractions and increased amount in the flow-through. The remaining Sam50 present in the elution fractions may be associated in the complex through the Sam35 residues that were not disrupted (S95, T92, T112) and the smaller Sam50-

Sam37 interface. The N-terminal truncation of Sam35 did not substantially alter the stoichiometry of Sam35 and Sam37, as demonstrated by the pulldown assay for the Sam35+Sam37 complex (Supplementary Figure 13C and D). The extensive interface between these two subunits does not include the N-terminal residues of Sam35, which is supported by the unchanged stoichiometry between Sam35 and Sam37 in these coexpressions.

Please note that while we agree that activity assays would be informative, the structure we solved is from a thermophilic fungus where no such assays exist, and the current COVID-19 restrictions prevent us from establishing such an assay in a timely manner. Where possible, we refer to assays performed in *S. cerevisiae* with the knowledge that no structures exist for any subunit from this organism and sequence similarities between *S. cerevisiae* and *M. thermophila* are limited (see supplementary figures 9, 10, 11 for comparisons).

(ii) residues of Sam37 that were suggested to interact with the POTRA domain of Sam50 should be mutated. Then, the impact on the structure of the complex, the association of Sam37 with the other two subunits (BN-PAGE), as well as the function of Mas37 (complementation of yeast strains lacking MAS37) should be investigated.

It was a great suggestion to test the interaction between Sam37 and the POTRA domain of Sam50. Indeed, we deleted the entire Sam50 POTRA domain to disrupt this interaction in our new cryoEM structure. The result showed the assembly of the SAM complex was not affected, with Sam35 and Sam37 assembling on Sam50 in the same way as with full-length Sam50. We note that the residues involved in this interaction show low sequence conservation (and in fact, in *S. cerevisiae* is not predicted to have a TM helix in Sam37). Therefore, we suggest this interaction between Sam37 and the POTRA domain is not essential for the assembly of the SAM complex. In the revised manuscript, we added a supplementary figure (Supplementary Fig. 14) to show the cryoEM reconstruction of the SAM complex without the POTRA domain and added the following sentence.

Page 12:

In the detergent structure, the Sam37 linker between transmembrane domains interacts with the Sam50 POTRA domain, but neither of these regions has high sequence conservation between species (Supplementary Fig. 6D). *The assembly of the SAM complex was maintained when the whole POTRA domain was truncated (Δ 1-135) (Supplementary Fig. 14), suggesting that this interaction between Sam37 and the POTRA domain is not essential for the assembly of the SAM complex.*

(iii) site-specific cross-linking assays can support the involvement of specific residues in protein-protein interactions.

In its current form, this contribution does not provide much new insight into the

mechanism of membrane assembly of β -barrel proteins. The lateral gate was already suggested based on the modelling of Sam50 on the structure of BamA (ref. 19) and biochemical assays (ref. 24).

This is another good suggestion, and one that we will certainly attempt in the future. Right now, crosslinking studies must be done in *S. cerevisiae* due to lack of an established assay for *M. thermophila*. As the reviewer points out, substantial crosslinking analysis was performed by Hohr et al. (ref. 24) that characterized the Sam50 lateral gate and also interactions between extracellular loop 6 and incoming substrate.

2. The modelling of the SAM-TOM supercomplex is based on very limited data and neglects the presence of the receptors Tom20 and Tom70 in native membranes. Thus, the statement that the supercomplex 'has no major clashes between members of the SAM and TOM complexes' is not accurate because some Tom subunits were neglected. I suggest that the authors will either support their model with more experimental data (biochemical or structural) or will remove this part.

This is a reasonable request, and we were in the process of collecting biochemical data to support the SAM-TOM model when the NIH was shut down in early March (we remain closed until at least May 31). Since we can't finish these experiments now, we will remove the model from this manuscript and validate it in the future. We removed the following sentence from the abstract:

The SAM complex structure suggests how it interacts with other mitochondrial outer membrane proteins to create supercomplexes.

Minor points:

1. Lines 82-3: This statement is not completely correct since newly synthesized Sam50 molecules are probably also substrates of pre-existing SAM complexes.

We corrected this statement on page 4, line 84:

Therefore, it appears that Sam50 substrates (other than Sam50 itself) may use a single folding and insertion mechanism.

2. An SDS-PAGE of the purified SAM complex should be shown to evaluate the purity of the complex. In addition, the size exclusion behavior of the purified complex should be presented.

This is a good point and we have included the requested data in supplementary figure 2.

3. The authors failed to cite the article of Klein et al. (JCB 2012) who presented a low-resolution structure of the TOB complex from *N. crassa* and reported a stoichiometry of 1:1:1 among the complex components.

We apologize for the oversight. The article is now cited appropriately in the first paragraph of the introduction (line 46).

4. The comparison of the structure of Sam50 to that of BamA takes a major part of this contribution and does not necessarily add mechanistic insights. I would suggest shortening this part.

We shortened the section on BamA and Sam50 comparisons by focusing primarily on differences at the lateral gate, and removing most of the comparison of loop 6 in the two structures.

Reviewer #2 (Remarks to the Author):

Mitochondrial β -barrel proteins play important roles in the outer membrane of mitochondria. After being synthesized in the cytosol, these proteins are imported across the mitochondrial outer membrane as unfolded precursor proteins, and then interact with the Sorting and Assembly Machinery (SAM) complex, which mediates the folding and membrane insertion of β -barrel proteins. The SMA complex contains one copy of three proteins: Sam50, Sam35 and Sam37. Sam50 is a 16-stranded β -barrel protein, and the other two proteins are located on the cytoplasmic side of the mitochondrial outer membrane. In this manuscript "Structural insight into mitochondrial β -barrel outer membrane protein biogenesis", the authors determined the cryo-EM structures of the SAM complex in lipid nanodiscs and in detergents at 3-4 Å resolutions. The SAM complex in nanodisc is monomeric, whereas it forms various dimers in the GDN detergent. Their structural analysis revealed the details of interactions between SAM components, suggested conformational dynamics of Sam50 β -barrel, and compared the structures of Sam50 and BamA, another member in the same Omp85 family. Cryo-EM work was nicely done, and the manuscript is well written. Their findings are interesting to the researchers in the related fields. However, a few important issues should be addressed before publication.

(1) It is surprising to find absolutely no biochemical data in the manuscript. SDS-PAGE and gel-filtration profiles are essential to evaluate the protein homogeneity. This is particularly relevant to the apparent discrepancy of oligomer states of the SAM complex in nanodiscs and GDN detergent. The authors should show gel-filtration profiles for the SAM complex in LMNG, GDN and nanodiscs.

Good point. The requested data are now shown in supplementary figure 2.

(2) Are there different oligomer states for SAM in nanodiscs, LMNG, or GDN? If so, are the distributions of different oligomers similar in these samples? Did GDN particularly induce dimer formation? Have the authors performed structural analysis on other oligomers of SAM in nanodisc or on SAM in LMNG?

We have conducted one analytical ultracentrifugation experiment of the SAM complex in LMNG; see Appendix 2, figures 1 and 2 at the end of this response letter. In this experiment, the major peak (78% of the absorbance) was composed of a 300±50kDa species that suggests a 2:2:2 ratio of Sam50:Sam35:Sam37. There were also small amounts of smaller and faster sedimenting species present in this sample.

We have not yet completed AUC with SAM complex purified in GDN or incorporated into lipid nanodiscs. Unfortunately, these experiments cannot be easily conducted at this point as the NIH is shut down until at least May 31 due to COVID-19. However, based on the cryoEM data we anticipate similar results of the SAM complex in GDN to the LMNG AUC experiment.

The different oligomeric states for SAM were observed during AUC of the sample in LMNG and cryoEM data analysis of the sample in GDN. The 2D and 3D classifications of cryoEM images of the sample in lipid nanodiscs showed only the monomeric state (as demonstrated in Supplementary Figure 5). We did not succeed in obtaining cryoEM images usable for data analysis from the sample in LMNG because lots of aggregates formed in the cryoEM grids, but from our AUC data we speculated that dimers may have formed during LMNG extraction. We noticed a significant sample loss during lipid nanodisc reconstitution from LMNG and therefore speculated that dimers were not favored for incorporation into lipid nanodiscs.

(3) In the section of “Opening of the Sam50 lateral gate observed in SAM complex dimers”, it is interesting and reasonable to speculate the β -strand interaction between substrate protein and Sam50. However, the conformational variation between SAM structures in nanodisc and in the GDN detergent should be treated with great caution. The possible effects from particular detergents on Sam50 structure and open/close β -barrel conformations should be discussed.

Yes, this is an important point and the text has been revised as follows, with additional references:

3D reconstructions for the SAM complex in GDN showed a mixture of monomers and three predominant dimer conformations (Supplementary Fig. 5). Among the dimer conformations, substantial differences were observed for the Sam50 β -barrel. *Before comparing these conformations, it is important to note that detergents have the potential to greatly distort membrane protein structures, while lipid environments such as nanodiscs appear to preserve a native or native-like conformation*(see for example refs:

Ward A, Reyes CL, Yu J, Roth CB, Chang G. 2007. Flexibility in the ABC transporter MsbA: alternating access with a twist. PNAS 104:19005–19010; Mi W, Li Y, Yoon SH, Ernst RK, Walz T, Liao M. 2017. Structural basis of MsbA-mediated lipopolysaccharide transport. Nature 549:233–237.

With these considerations in mind, we note that in dimer 1 as previously discussed, the Sam50 β -barrel is partially closed.

(4) The modeling of TOM-SAM supercomplex is interesting, but highly

speculative without the support of more structural and biochemical data. Therefore, current Fig. 6 seems to be more appropriate as a supplementary figure.

Since it is speculative and reviewer 1 also had questions about it, we decided to remove it from the manuscript. We removed the following sentence from the abstract:

The SAM complex structure suggests how it interacts with other mitochondrial outer membrane proteins to create supercomplexes.

(5) For a general reader, a schematic at the beginning of the manuscript would be helpful, to illustrate the structural features in SAM, and to compare SAM and BAM. This should indicate various domains including all β -strands, loop names, and Sam37 TM and linker. This can also be done in Fig. 1, when the SAM structure is first presented.

To address these questions, we added a schematic as new figure 1 showing proteins and pathways in the mitochondrial outer membrane. We also added a cartoon comparison of BAM and SAM in supplementary figure 1 and created another new supplementary figure 4 to show labeling of secondary structure elements. To give complete secondary structure information, we also added supplementary figures (10 and 11) showing sequence alignments for Sam35 and Sam37 with their secondary structure elements shown above the sequences.

(6) Line 160. "... the most prevalent dimer (dimer 1)". This seems a wrong statement according to Suppl. Fig. 2. Dimer 3 appears to have more particles in both 1st round 3D classification and in the final refinement step. What is the reason for the authors to focus on dimer 1? What would be the results if the same extensive image processing procedure for dimer 1 is applied on dimer 2 and dimer 3?

We thank the reviewer for pointing out this inaccurate statement. Because the current algorithms for image classification in single particle analysis integrate particles into limited numbers of classes, the particle number for each class may not correctly represent the size of population when there are strong flexibilities or heterogeneities, which we think is the case for dimer 3. The major body of particles in this class were not selected for the final 3D reconstruction, suggesting that this class contained a relatively heterogeneous population. The reasons we focused on dimer 1 were two-fold: 1) The SAM complex in dimer 1 showed the same conformation as that found in the monomer structure of the SAM in lipid nanodiscs. 2) Dimer 1 had the most homogeneous particles that achieved a high-resolution reconstruction. There were multiple reasons we did not extend the same extensive image processing procedure on dimer 2 and dimer 3: 1) Comparisons among these three structures showed nearly identical structures of Sam35 and Sam37 as well as the interfaces between Sam50, Sam35, and Sam37. Improvement of resolution for dimer 2 or dimer 3 would not provide new information. 2) Dimer 2 and dimer 3 were at a resolution lower than that of dimer 1 due to a smaller

population of particles and/or a higher heterogeneity, therefore particle polishing and symmetry expansion would not be as effective as those for a large number of homogeneous particles such as dimer 1. 3) Particle polishing and symmetry expansion are computationally expensive. Due to these reasons and our limited access to computational resources, we did not perform the same extensive data analysis for dimer 2 and dimer 3.

We modified Line 162 to a more accurate statement: “However, we were able to determine a structure of *the dimer with the most homogeneous conformation (dimer 1)* at 3.2 Å resolution from 117,339 particles (Fig. 3A).”

(7) In several parts of the manuscript, the analysis/presentation would make more sense if the points in the text are indicated in figures.

Fig.1. Indicate aa 339-353 (related to text in line 140) and the presumable position of TM helices of Sam37, and POTRA domain of Sam50.

We have included a new supplementary figure 4 to show secondary structure elements and selected residues for Sam50, Sam35, and Sam37.

Line 228-242 has detailed discussion about interactions between conserved residues. But not a single residue was indicated in Supplementary Fig. 6, even though the levels of conservation were colored.

Residues are now labeled as requested.

Line 258. Where is this conserved glycine in the structure?

See above.

Line 144. “At low resolution ...” and Line 145 “at high resolution” are perhaps better to change to “When filtered to low/high resolution...”. In addition, these maps filtered to different resolutions should be shown to support the claims.

We have now added filtered maps to new Supplementary Fig. 1E and modified Line 145 as “*When the map is filtered to low resolution, the first of two predicted transmembrane α -helices in Sam37 is visible; however, it is not visible when filtered to high resolution (Supplementary Fig. 3E).*”

Other minor points:

Can the authors explicitly list all available structures of Omp85 members in Introduction?

We added the requested references as follows (line 76):

Sam50 and BamA are evolutionarily related, and both are members of the Omp85 superfamily^{9,23}. Although no structures had previously been determined for any of the SAM components, *structures exist for Omp85 family members BamA, TamA, and FhaC. (all structures now cited).*

Throughout the text and methods, many “Fig. S#” should be changed to “Supplementary Fig. #”.

Done.

Fig. 4. The right panels do not exactly match the left panels. It would be helpful to label certain residues in the right panels to orient the readers.

Done.

Suppl. Fig. 1C. The final step is using cryoSPARC with non-uniform refinement. Can the authors show the results from (1) CryoSPARC without non-uniform refinement, and (2) Relion auto-refine? This would be interesting and informative for the users of these programs.

We have now added the results from cryoSPARC homogeneous refinement and RELION auto-refinement in Supplementary Fig. 3C.

Suppl. Fig. 1E and 3B. Among these FSCs, only the 3rd and 5th are essential. However, these figures are missing two validation FSCs: [model refined against half map1] vs. [half map1], and [model refined against half map1] vs. [half map2]. The consistency between these two FSCs would prove no over-refinement of the model.

The FSCs were generated and are shown in supplementary figures 3 and 6.

Suppl. Fig. 3D. What are the sigma levels for display?

Two density maps are displayed at 1.0 sigma level in Supplementary Fig. 3D. The sigma values were calculated locally using a subregion slightly larger than the displayed region. The sigma level for display is now indicated in the figure caption.

Suppl. Table 3. What is the criteria/threshold for assigning a residue as “conserved”?

Residues are absolutely conserved across 7 species, identified by sequence alignments. This statement has been added to the legend in Supplementary Table 3.

Appendix 1. *In silico* simulations of selected mutants at interfaces.

		Values are changes relative to wild type (in blue bold)				
		Interface area	Delta G	Binding E	H-bonds	Salt bridges
Sam35 wt	to Sam50	1919.1	-16.2	-26.4	18	6
N35A		-33.7	-1.6	-0.7	-2	0
N35A, E36A		-73.6	-2.6	-0.5	-3	-2
N35A, E36A, E39A		-73.6	-2.7	0.3	-5	-2
N35A, E36A, R108A		-58	-2.7	0.8	-3	-6
E36A		-25.1	-0.9	0.2	-1	-2
E36A, E39A		-39.9	-1.1	1	-3	-2
E36A, E39A, R108A		-39.1	-1.2	2.3	-3	-6
E36A, R108A		-24.3	-1.1	1.6	-1	-6
Sam35 wt	to Sam37	1978.7	-16.8	-26.4	15	8
Y156A		-11.9	0	0.8	-2	0
Y156A, K221A		-39.6	-1	0.3	-3	0
Y156A, K221A, R241A		-89.3	-0.4	1.8	-5	-3
R152A		-34.2	-1.3	-1	0	-1
R152A, D229A		-41.1	-1.6	0.2	0	-5
Sam50 wt	to Sam35	1919.1	-16.2	-26.4	18	6
N431A		-0.6	-0.4	0	-1	0
N431A, S432A		-4	-0.8	0	-2	0
N431A, S432A, E497A		3.6	-1	0.3	-3	0
D386A		-3.1	-0.7	0.8	0	-4
D386A, N431A		-3.7	-1.1	0.8	-1	-4
D386A, N431A, K433A		-42.1	-1.8	0.9	-1	-6
D386A, S432A, K433A		-37.2	-1.6	1.1	-1	-6
D386A, K433A		-33.8	-1.2	1	0	-6
Sam50 wt	to Sam37	945.7	-9.2	-13.2	9	0
R59A		-52.3	-1	-0.5	-1	0
R59A, R137A		-83.2	-2	-0.7	-3	0
Sam37 wt	to Sam35	1978.7	-16.8	-26.4	15	8
S69A		-4	-0.4	0	-1	0
S69A, D111A		-8.1	-1.4	-0.2	-2	-1
S69A, D111A, R168A		-64.4	-1.9	-0.2	-3	-1
S69A, D111A, R363A		-40.3	-3.3	-0.6	-2	-5
D111A		-4.1	-0.9	-0.2	-1	-1
D111A, R363A		-36.3	-2.8	-0.6	-1	-5
Sam37 wt	to Sam50	945.7	-9.2	-13.2	9	0
Q418A		-20.4	-0.7	-0.2	-1	0

Appendix 2. AUC data for SAM complex purified from LMNG

Appendix Figure 1. Analytical Ultracentrifugation data for the SAM complex purified in LMNG.

Top panel displays absorbance, bottom panel interference. Major sedimenting species (78% of the absorbance) at 12.98S calculated to be 300 ± 50 kDa protein, suggests a 2:2:2 ratio of Sam50:Sam35:Sam37. Small amounts of smaller and faster sedimenting species present, free LMNG detergent micelles present at 3-4S.

AUC experiment and analysis conducted by Rodolfo Ghirlando (NIDDK, NIH)

All calculations:

Total= 430 ± 70 kDa

Protein= 300 ± 50 kDa

LMNG= 130 ± 6 kDa

Appendix Figure 2. Purification of SAM complex in LMNG for AUC experiment. SAM complex containing Sam37 with N-terminal Twin-Strep tag was solubilized in LMNG, purified by **(A)** strep affinity and **(B)** Superose6 size exclusion chromatography. **(C)** SDS-PAGE of purified SAM complex sample left at room temperature for 8 hours.

REVIEWERS' COMMENTS:

Reviewer #1 (Remarks to the Author):

The authors addressed most of my comments regarding the original version.

In those cases where they did not perform the suggested experiments, they provided reasonable explanations why such experiments are beyond the scope of the current contribution or cannot be performed due to Corona-related work restrictions.

Reviewer #2 (Remarks to the Author):

The authors have addressed all the issues raised.